# DIFFUSION LANGUAGE MODELS KNOW THE ANSWER BEFORE DECODING

**Pengxiang Li[1],[\*] Yefan Zhou[2],[\*] Dilxat Muhtar[5],[6],[7], Lu Yin[3], Shilin Yan, Li Shen[4]**
**Soroush Vosoughi[2], Shiwei Liu[5],[6],[7]**
[1]The Hong Kong Polytechnic University    [2]Dartmouth College
[3]University of Surrey    [4]Sun Yat-sen University
[5]ELLIS Institute Tübingen    [6]MPI for Intelligent Systems    [7]Tübingen AI Center

## ABSTRACT

Diffusion language models (DLMs) have recently emerged as an alternative to autoregressive approaches, offering parallel sequence generation and flexible token orders. However, their inference remains slower than that of autoregressive models, primarily due to the cost of bidirectional attention and the large number of refinement steps required for high-quality outputs. In this work, we highlight and leverage an overlooked property of DLMs—**early answer convergence**: in many cases, the correct answer can be internally identified by half steps before the final decoding step, under both semi-autoregressive and random remasking schedules. For example, on GSM8K and MMLU, up to 97% and 99% of instances, respectively, can be decoded correctly using only half of the refinement steps. Building on this observation, we introduce **Prophet**, a training-free fast decoding paradigm that enables **early commit decoding**. Specifically, Prophet dynamically decides whether to continue refinement or to go "all-in" (i.e. decode all remaining tokens in one step), using the confidence gap between the top-2 prediction candidates as the criterion. It integrates seamlessly into existing DLM implementations, incurs negligible overhead, and requires no additional training. Empirical evaluations on LLaDA-8B and Dream-7B across multiple tasks show that Prophet reduces the number of decoding steps by up to 3.4× while preserving high generation quality, and yields additional speedups when combined with existing acceleration methods. These results recast DLM decoding as a problem of *when to stop sampling*, and demonstrate that early answer convergence provides a simple yet powerful mechanism for accelerating DLMs on reasoning, code, and planning tasks with identifiable answer regions. Our code is available at `https://github.com/pixeli99/Prophet`.

## 1 INTRODUCTION

Along with the rapid evolution of diffusion models in various domains (Ho et al., 2020; Nichol & Dhariwal, 2021; Ramesh et al., 2021; Saharia et al., 2022; Jing et al., 2022), Diffusion language models (DLMs) have emerged as a compelling and competitively efficient alternative to autoregressive (AR) models for sequence generation (Austin et al., 2021a; Lou et al., 2023; Shi et al., 2024; Sahoo et al., 2024; Nie et al., 2025; Gong et al., 2024; Ye et al., 2025). Primary strengths of DLMs over AR models include, but are not limited to, efficient parallel decoding and flexible generation orders. More specifically, DLMs decode all tokens in parallel through iterative denoising and re-masking steps. The remaining tokens are typically refined with low-confidence predictions over successive rounds (Nie et al., 2025).

Despite the potential speedup of DLMs, the inference speed of DLMs is slower than that of AR models in practice, due to the lack of KV cache mechanisms and the significant performance degradation associated with fast parallel decoding (Israel et al., 2025a). Recent efforts have proposed excellent algorithms to enable the KV cache (Ma et al., 2025a; Liu et al., 2025a; Wu et al., 2025) and improve the performance of parallel decoding (Wu et al., 2025; Wei et al., 2025a; Hu et al., 2025).

---

[\*]Equal contribution.

In this paper, we aim to accelerate the inference of DLMs from a different perspective, motivated by an overlooked yet powerful phenomenon of DLMs—**early answer convergence**. Through extensive analysis, we observed that: *a strikingly high proportion of samples can be correctly decoded during the early phase of decoding for both semi-autoregressive remasking and random remasking.* This trend becomes more significant for random remasking. For example, on GSM8K and MMLU, up to 97% and 99% of instances, respectively, can be decoded correctly using only half of the refinement steps.

Motivated by this finding, we introduce **Prophet**, a training-free fast decoding strategy designed to capitalize on early answer convergence. Prophet continuously monitors the confidence gap between the top-2 answer candidates throughout the decoding trajectory, and opportunistically decides whether it is safe to decode all remaining tokens at once. By doing so, Prophet achieves substantial inference speedup (up to $3.4\times$) while maintaining high generation quality. Our contributions are threefold:

- **Empirical observations of early answer convergence:** We demonstrate that a strikingly high proportion of samples (up to 99%) can be correctly decoded during the early phase of decoding for both semi-autoregressive remasking and random remasking. This underscores a fundamental redundancy in conventional full-length slow decoding.

- **A fast decoding paradigm enabling early commit decoding:** We propose Prophet, which evaluates at each step whether the remaining answer is accurate enough to be finalized immediately, which we call Early Commit Decoding. We find that the confidence gap between the top-2 answer candidates serves as an effective metric to determine the right time for early commit decoding. Using this metric, Prophet dynamically decides between continued refinement and immediate answer emission.

- **Substantial speedup gains with high-quality generation:** Experiments across diverse reasoning, code, and planning benchmarks with identifiable answer regions reveal that Prophet delivers up to $3.4\times$ reduction in decoding steps. Crucially, this acceleration incurs negligible degradation in accuracy-affirming that early commit decoding is not just computationally efficient but also semantically reliable for DLMs.

## 2 RELATED WORK

### 2.1 DIFFUSION LARGE LANGUAGE MODEL

The idea of adapting diffusion processes to discrete domains traces back to the pioneering works of Sohl-Dickstein et al. (2015); Hoogeboom et al. (2021). A general probabilistic framework was later developed in D3PM (Austin et al., 2021a), which modeled the forward process as a discrete-state Markov chain progressively adding noise to the clean input sequence over time steps. The reverse process is parameterized to predict the clean text sequence based on the current noisy input by maximizing the Evidence Lower Bound (ELBO). This perspective was subsequently extended to the continuous-time setting. Campbell et al. (2022) reinterpreted the discrete chain within a continuous-time Markov chain (CTMC) formulation. An alternative line of work, SEDD (Lou et al., 2023), focused on directly estimating likelihood ratios and introduced a denoising score entropy criterion for training. Recent analyses in MDLM (Shi et al., 2024; Sahoo et al., 2024; Zheng et al., 2024) and RADD (Ou et al., 2024) demonstrate that multiple parameterizations of MDMs are in fact equivalent.

Motivated by these groundbreaking breakthroughs, practitioners have successfully built product-level DLMs. Notable examples include commercial releases such as Mercury (Labs et al., 2025), Gemini Diffusion (DeepMind, 2025), and Seed Diffusion (Song et al., 2025b), as well as open-source implementations including LLaDA (Nie et al., 2025) and Dream (Ye et al., 2025). However, DLMs face an efficiency-accuracy tradeoff that limits their practical advantages. While DLMs can theoretically decode multiple tokens per denoising step, increasing the number of simultaneously decoded tokens results in degraded quality. Conversely, decoding a limited number of tokens per denoising step leads to high inference latency compared to AR models, as DLMs cannot naively leverage key-value (KV) caching or other advanced optimization techniques due to their bidirectional nature.

## 2.2 Acceleration Methods for Diffusion Language Models

Recent efforts to accelerate DLM inference while maintaining quality span several directions.

**KV Cache optimization.** One line of work exploits the high similarity between hidden states across consecutive denoising steps to enable approximate caching (Ma et al., 2025b; Liu et al., 2025b; Hu et al., 2025). A related strategy restructures the denoising process in a semi-autoregressive or block-autoregressive manner, caching states from previous context or blocks, optionally with periodic cache refreshing (Wu et al., 2026; Arriola et al., 2025; Wang et al., 2025b; Song et al., 2025a).

**Token pruning.** A second direction reduces attention cost by pruning redundant tokens (Xiao et al., 2026; Chen et al., 2026a). For example, DPad (Chen et al., 2026a) is a training-free method that treats suffix tokens as a computational scratchpad and prunes distant ones prior to computation.

**Sampling and decoding optimization.** A third direction optimizes the sampling or decoding process, either by increasing the number of tokens decoded per step or by reducing the total number of denoising steps. Training-based approaches include reinforcement learning (Song et al., 2025b) and distillation (Wang et al., 2025a; Chen et al., 2026b). Our work is most closely related to the training-free approaches in this direction. Fast-dLLM (Wu et al., 2026) combines approximate block-wise KV caching with confidence-aware parallel decoding that commits tokens exceeding a confidence threshold. SlowFast (Wei et al., 2025a) proposes a dynamic scheme that alternates between a cautious exploratory phase and an accelerated phase guided by three principles: certainty, convergence, and position. WINO (Song et al., 2025a) introduces a revokable draft-and-verify process that aggressively drafts multiple tokens and remasks low-confidence ones via an auxiliary shadow block. Other relevant work studies better criteria and metrics for inference-time decisions, including analyses of denoising dynamics (Wei et al., 2025b; Huang & Tang, 2025), alignment with small autoregressive models (Israel et al., 2025b), and speculative decoding using the DLM itself as a draft model (Agrawal et al., 2025).

**Positioning of our study.** The primary contribution of this work is to identify, formalize, and validate Early Answer Convergence, a fundamental property of the DLM denoising trajectory. The proposed method Prophet operationalizes this property by treating decoding as an optimal stopping problem over the answer region, deciding when to terminate further refinement based on answer confidence. Unlike methods such as Fast-dLLM that optimize the cost per step, Prophet reduces the total number of steps required. The two approaches are therefore orthogonal and can be combined for multiplicative speedups, as supported empirically in later sections. We note that Early Answer Convergence has been independently discovered in concurrent work (Wang et al., 2025a); however, their focus is on averaging predictions across time steps to improve accuracy, whereas we develop an early commit decoding method that reduces computational cost while maintaining quality.

## 3 Preliminary

### 3.1 Background on Diffusion Language Models

Concretely, let $x_0 \sim p_{\text{data}}(x_0)$ be a clean input sequence. At an intermediate noise level $t \in [0, T]$, we denote by $x_t$ the corrupted version obtained after applying a masking procedure to a subset of its tokens.

**Forward process.** The corruption mechanism can be expressed as a Markov chain

$$q(x_{1:T} \mid x_0) = \prod_{t=1}^{T} q(x_t \mid x_{t-1}), \tag{1}$$

which gradually transforms the original sample $x_0$ into a maximally degraded representation $x_T$. At each step, additional noise is injected, so that the sequence becomes progressively more masked as $t$ increases.

While the forward process in Eq.(1) is straightforward, its exact reversal is typically inefficient because it unmasks only one position per step (Campbell et al., 2022; Lou et al., 2023). To accelerate generation, a common remedy is to use the $\tau$-*leaping* approximation (Gillespie, 2001), which enables multiple masked positions to be recovered simultaneously. Concretely, transitioning from

corruption level $t$ to an earlier level $s < t$ can be approximated as

$$q_{s|t} = \prod_{i=1}^{n} q_{s|t}(x_s^i \mid x_t), \quad q_{s|t}(x_s^i \mid x_t) = \begin{cases} 1, & x_t^i \neq [\text{MASK}], \ x_s^i = x_t^i, \\ \frac{s}{t}, & x_t^i = [\text{MASK}], \ x_s^i = [\text{MASK}], \\ \frac{t-s}{t} \, q_{0|t}(x_s^i \mid x_t), & x_t^i = [\text{MASK}], \ x_s^i \neq [\text{MASK}]. \end{cases} \tag{2}$$

Here, $q_{0|t}(x_s^i \mid x_t)$ is a predictive distribution over the vocabulary, supplied by the model itself, whenever a masked location is to be unmasked. In conditional generation (e.g., producing a response $x_0$ given a prompt $p$), this predictive distribution additionally depends on $p$, i.e., $q_{0|t}(x_s^i \mid x_t, p)$.

**Reverse generation.** To synthesize text, one needs to approximate the reverse dynamics. The generative model is parameterized as

$$p_\theta(x_{0:T}) \;=\; p_\theta(x_T) \prod_{t=1}^{T} p_\theta(x_{t-1} \mid x_t) \;=\; \prod_{t=1}^{T} q(x_{t-1} \mid x_0) \, p_\theta(x_0 \mid x_t). \tag{3}$$

This reverse process naturally decomposes into two complementary components. (1) Prediction step. The model $p_\theta(x_0 \mid x_t)$ attempts to reconstruct a clean sequence from the corrupted input at level $t$. We denote the predicted sequence after this step by $x_0^t$, i.e. $x_0^t = p_\theta(x_0 \mid x_t)$. (2) Remasking step. Once a candidate reconstruction $x_0^t$ is obtained, the forward noising mechanism is reapplied in order to produce a partially corrupted sequence $x_{t-1}$ that is less noisy than $x_t$. This "remasking" can be implemented in various ways, such as masking tokens uniformly at random or selectively masking low-confidence positions (Nie et al., 2025). Through the interplay of these two steps—prediction and remasking—the model iteratively refines an initially noisy sequence into a coherent text output.

## 3.2 Early Answer Convergency

In this section, we investigate the early emergence of correct answers in DLMs. We conduct a comprehensive analysis using LLaDA-8B (Nie et al., 2025) on two widely used benchmarks: GSM8K (Cobbe et al., 2021) and MMLU (Hendrycks et al., 2021). Specifically, we examine the decoding dynamics, that is, how the top-1 predicted token evolves across positions at each decoding step, and report the percentage of the full decoding process at which the top-1 predicted tokens first match the ground truth answer tokens. In this study, we only consider samples where the final output contains the ground truth answer.

For low confidence remasking, we set Answer length at 256 and Block length at 32 for GSM8K, and Answer length at 128 and Block length to 128 for MMLU. For random remasking, we set Answer length at 256 and Block length at 256 for GSM8K, and Answer length at 128 and Block length at 128 for MMLU. We present the results of GSM8K in Figure 1, and the MMLU results in Appendix B.

**I. A high proportion of samples can be correctly decoded during the early phase of decoding.** Figure 1a demonstrates that when remasking with the low-confidence strategy, 24.2% samples are already correctly predicted in the first half steps, and 7.9% samples can be correctly decoded in the first 25% steps. These two numbers will be further largely boosted to 97.2% and 88.5%, when shifted to random remasking as shown in Figure 1c.

**II. Our suffix prompt further amplifies the early emergence of correct answers.** Adding the suffix prompt "Answer:" significantly improves early decoding. With low confidence remasking, the proportion of correct samples emerging by the 25% step rises from 7.9% to 59.7%, and by the 50% step from 24.2% to 75.8% (Figure 1b). Similarly, under random remasking, the 25% step proportion increases from 88.5% to 94.6%. We clarify that the suffix prompt acts as a semantic anchor rather than introducing oracle information. Since DLMs generate bidirectionally, this anchor explicitly conditions the model to locate the solution in the designated region, reducing the search space and accelerating convergence.

**III. Decoding dynamics of chain-of-thought tokens.** We further examine the decoding dynamics of chain-of-thought tokens in addition to answer tokens, as shown in Figure 2. First, most non-answer tokens fluctuate frequently before being finalized. Second, answer tokens change far less often and tend to stabilize earlier, remaining unchanged for the rest of the decoding process.

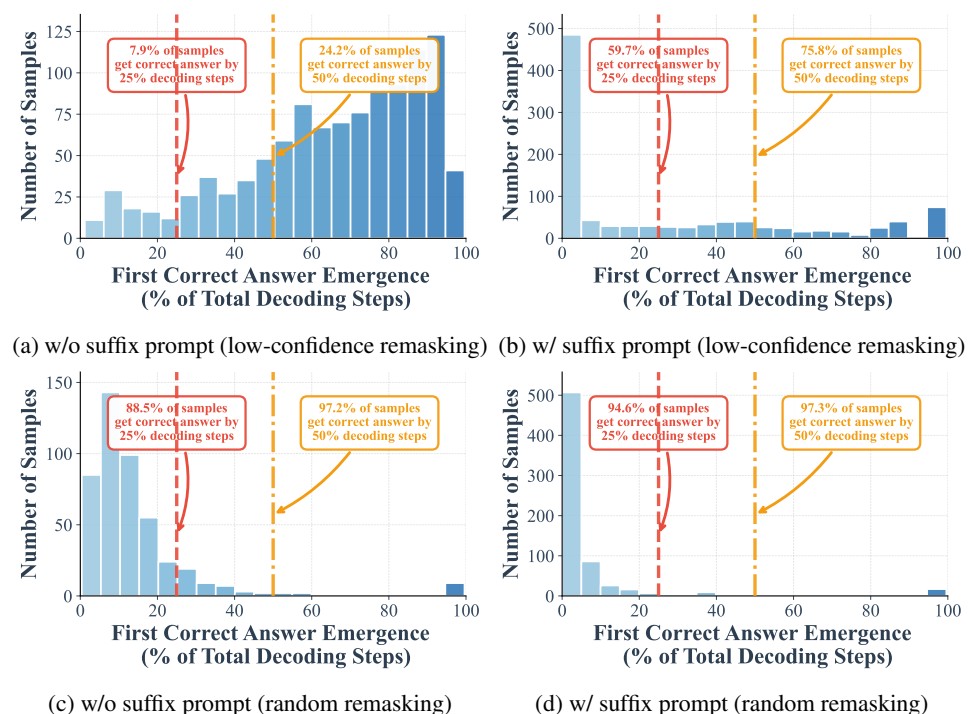

Figure 1: **Distribution of early correct answer detection during decoding process.**. Histograms show when correct answers first emerge during diffusion decoding, measured as percentage of total decoding steps, using `LLaDA 8B` on `GSM8K`. Red and orange dashed lines indicate 50% and 70% completion thresholds, with corresponding statistics showing substantial early convergence. Suffix prompting (b,d) dramatically accelerates convergence compared to standard prompting (a,c). This early convergence pattern demonstrates that correct answer tokens stabilize as top-1 candidates well before full decoding.

## 4 METHODOLOGY

Based on the above findings, we introduce **Prophet**, a training-free fast decoding algorithm designed to accelerate the generation phase of DLMs. Prophet by committing to all remaining tokens in one shot and predicting answers as soon as the model's predictions have stabilized, which we call Early Commit Decoding. Unlike conventional fixed-step decoding, Prophet actively monitors the model's certainty at each step to make an informed, on-the-fly decision about when to finalize the generation.

**Confidence Gap as a Convergence Metric.** The core mechanism of Prophet is the **Confidence Gap**, a simple yet effective metric for quantifying the model's conviction for a given token. Let $N_{\text{gen}}$ be the total generation length. In semi-autoregressive decoding, tokens are generated in blocks of size $N_{\text{block}}$. Prophet focuses on monitoring the *Answer Region* $\mathcal{A}$ of length $N_{\text{ans}}$ within the current generation window. At any decoding step $t$, the DLM produces a logit matrix $\mathbf{L}_t \in \mathbb{R}^{N \times |\mathcal{V}|}$, where $N$ is the sequence length and $|\mathcal{V}|$ is the vocabulary size. For each position $i$, we identify the highest logit value, $L_{t,i}^{(1)}$, and the second-highest, $L_{t,i}^{(2)}$. The confidence gap $g_{t,i}$ is defined as their difference:

$$g_{t,i} = L_{t,i}^{(1)} - L_{t,i}^{(2)}. \tag{4}$$

This value serves as a robust indicator of predictive certainty. A large probability gap signals that the prediction has likely converged, with the top-ranked token clearly outweighing all others. We calculate the average confidence gap $\bar{g}_t$ exclusively over the answer region $\mathcal{A}$ (i.e., $\bar{g}_t = \frac{1}{|\mathcal{A}|} \sum_{i \in \mathcal{A}} g_{t,i}$), rather than the entire sequence, to maximize sensitivity.

**Early Commit Decoding.** The decision of when to terminate the decoding loop can be framed as an optimal stopping problem. At each step, we must balance two competing costs: the **computa-**

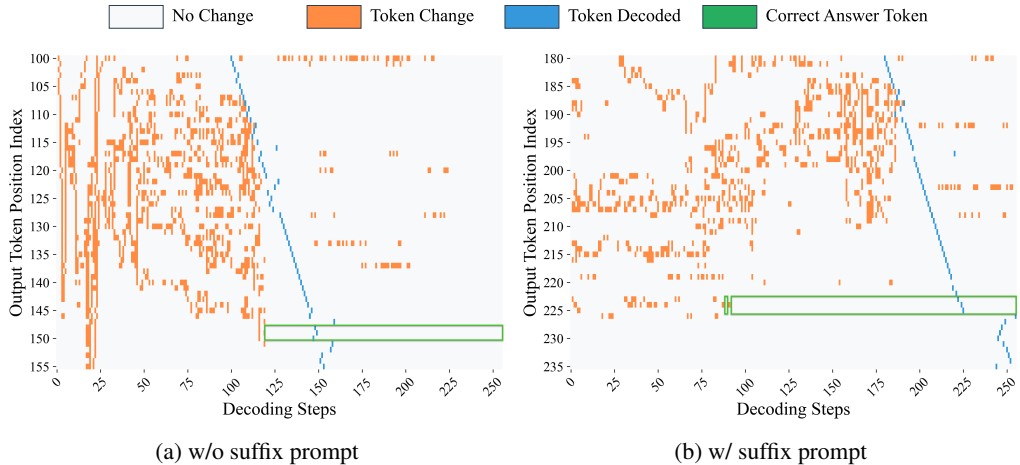

Figure 2: **Decoding dynamics across all positions based on maximum-probability predictions.** Heatmaps track how the top-1 token changes at each position, if it is decoded at the current step, over the course of decoding. (a) Without our suffix prompts, correct answer tokens reach maximum probability at step 119. (b) With our suffix prompts, this occurs earlier at step 88, showing that the model internally identifies correct answers well before the final output. Results are shown for `LLaDA 8B` solving problem index 700 from GSM8K under low-confidence decoding. Gray indicates positions where the top-1 prediction remains unchanged, orange marks positions where the prediction changes to a different token, blue denotes the step at which the corresponding y-axis position is actually decoded, and green box highlights the answer region where the correct answer remains stable as the top-1 token and can be safely decoded without further changes as the decoding process progresses.

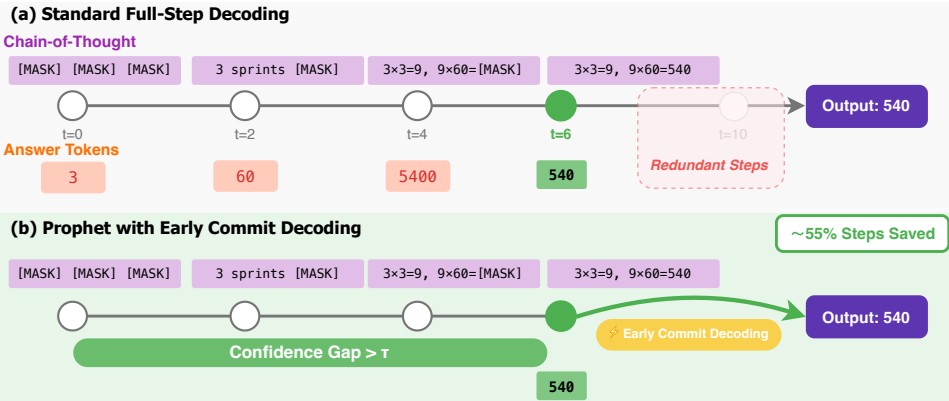

Figure 3: An illustration of the Prophet's early-commit-decoding mechanism. (a) Standard full-step decoding completes all predefined steps (e.g., 10 steps), incurring redundant computations after the answer has stabilized (at t=6). (b) Prophet dynamically monitors the model's confidence (the "Confidence Gap"). It triggers an early commit decoding as soon as the answer converges, saving a significant portion of the decoding steps (in this case, 55%) without compromising the output quality.

**tional cost** of performing additional refinement iterations versus the **risk of error** from a premature and potentially incorrect decision. The computational cost is a function of the remaining steps, whereas the risk of error is inversely correlated with the model's predictive certainty, for which the Confidence Gap serves as a robust proxy.

Prophet addresses this trade-off with an adaptive strategy that embodies a principle of **time-varying risk aversion**. Let denote $p = (T_{\max} - t)/T_{\max}$ as the decoding progress, where $T_{\max}$ is the total number of decoding steps, and $\tau(p)$ is the threshold for early commit decoding. In the early noisy

stages of decoding (when progress $p$ is small), the potential for significant improvement in prediction is high. Committing to an answer at this stage carries a high risk. Therefore, Prophet acts in a risk-averse manner, demanding an exceptionally high threshold ($\tau_{\text{high}}$) to justify an early commit decoding, ensuring that such a decision is unequivocally safe. As the decoding process matures (as $p$ increases), two things happen: the model's predictions stabilize, and the potential computational savings from stopping early diminish. Consequently, the cost of performing one more step becomes negligible compared to the benefit of finalizing the answer. Prophet thus becomes more risk-tolerant, requiring a progressively smaller threshold ($\tau_{\text{low}}$) to confirm convergence.

This dynamic risk-aversion policy is instantiated through our staged threshold function, which maps the abstract trade-off between inference speed and generation certainty onto a concrete decision rule:

$$\bar{g}_t \geq \tau(p), \quad \text{where} \quad \tau(p) = \begin{cases} \tau_{\text{high}} & \text{if } p < 0.33 \\ \tau_{\text{mid}} & \text{if } 0.33 \leq p < 0.67 \\ \tau_{\text{low}} & \text{if } p \geq 0.67 \end{cases} \tag{5}$$

Once the exit condition is satisfied at step $t^*$, the iterative loop is terminated. The final output is then constructed in a single parallel operation by filling any remaining [MASK] tokens with the argmax of the current logits $\mathbf{L}_{t^*}$.

**Algorithm Summary.** The complete Prophet decoding procedure is outlined in Algorithm 1. Integration of the confidence gap check adds negligible computational overhead to the standard DLM decoding loop. Prophet is model-agnostic, requires no retraining, and can be readily implemented as a wrapper around existing DLM inference code.

---

**Algorithm 1** Prophet: Early Commit Decoding for Diffusion Language Models
___
1: **Input:** Model $M_\theta$, prompt $\mathbf{x}_{\text{prompt}}$, max steps $T_{\text{max}}$, total generation length $N_{\text{gen}}$
2: **Input:** Threshold function $\tau(\cdot)$, answer region $\mathcal{A}$ (where $|\mathcal{A}| = N_{\text{ans}}$)
3: Initialize sequence $\mathbf{x}_T \leftarrow \text{concat}(\mathbf{x}_{\text{prompt}}, [\text{MASK}]^{N_{\text{gen}}})$
4: Let $\mathcal{M}_t$ be the set of masked positions at step $t$.
5: **for** $t = T_{\text{max}}, T_{\text{max}} - 1, \ldots, 1$ **do**
6:     Compute logits: $\mathbf{L}_t = M_\theta(\mathbf{x}_t)$
7:                                                           ▷ **Prophet's Early-Commit-Decoding Check**
8:     Calculate average confidence gap $\bar{g}_t$ over positions $\mathcal{A}$ using Eq. 4.
9:     Calculate progress: $p \leftarrow (T_{\text{max}} - t)/T_{\text{max}}$
10:     **if** $\bar{g}_t \geq \tau(p)$ **then**                                   ▷ Check condition from Eq. 5
11:         $\hat{\mathbf{x}}_0 \leftarrow \text{argmax}(\mathbf{L}_t, \dim = -1)$
12:         $\mathbf{x}_0 \leftarrow \mathbf{x}_t$. Fill positions in $\mathcal{M}_t$ with tokens from $\hat{\mathbf{x}}_0$.
13:         **Return** $\mathbf{x}_0$                                       ▷ Terminate and finalize
14:     **end if**
15:                                                     ▷ **Standard DLM Refinement Step**
16:     Determine tokens to unmask $\mathcal{U}_t \subseteq \mathcal{M}_t$ via a remasking strategy.
17:     $\hat{\mathbf{x}}_0 \leftarrow \text{argmax}(\mathbf{L}_t, \dim = -1)$
18:     Update $\mathbf{x}_{t-1} \leftarrow \mathbf{x}_t$, replacing tokens at positions $\mathcal{U}_t$ with those from $\hat{\mathbf{x}}_0$.
19: **end for**
20: **Return** $\mathbf{x}_0$                   ▷ Return result after full iterations if no early commit decoding

---

## 5 EXPERIMENTS

We evaluate Prophet on DLMs to validate two key hypotheses: first, that Prophet can preserve the performance of full-budget decoding while using substantially fewer denoising steps; second, that our adaptive approach provides more reliable acceleration than naive static baselines. We demonstrate that Prophet achieves notable computational savings with negligible quality degradation through comprehensive experiments across diverse benchmarks.

### 5.1 EXPERIMENTAL SETUP

We conduct experiments on two state-of-the-art diffusion language models: LLaDA-8B (Nie et al., 2025) and Dream-7B (Ye et al., 2025). For each model, we compare two decoding strategies: **Full**

Table 1: Benchmark results on LLaDA-8B-Instruct and Dream-7B-Instruct. We report Accuracy (%) for both Full-step decoding and Prophet. The numbers in parentheses indicate the **Accuracy Gain** ($\Delta$) compared to the baseline. Sudoku and Countdown are evaluated using 8-shot setting; all other benchmarks use zero-shot evaluation. Detailed configuration is listed in the Appendix C.

| Benchmark | LLaDA-8B | | | Dream-7B | | |
|---|---|---|---|---|---|---|
| | Full (%) | Prophet ($\Delta$) | Speedup | Full (%) | Prophet ($\Delta$) | Speedup |
| *General Tasks* | | | | | | |
| MMLU | 54.1 | 54.0 (-0.1) | 2.34× | 67.6 | 66.1 (-1.5) | 2.47× |
| ARC-C | 83.2 | 83.5 (+0.3) | 1.88× | 88.1 | 87.9 (-0.2) | 2.61× |
| Hellaswag | 68.7 | 70.9 (+2.2) | 2.14× | 81.2 | 81.9 (+0.7) | 2.55× |
| TruthfulQA | 34.4 | 46.1 (+11.7) | 2.31× | 55.6 | 53.2 (-2.4) | 1.83× |
| WinoGrande | 73.8 | 70.5 (-3.3) | 1.71× | 62.5 | 62.0 (-0.5) | 1.45× |
| PIQA | 80.9 | 81.9 (+1.0) | 1.98× | 86.1 | 86.6 (+0.5) | 2.29× |
| *Mathematics & Scientific* | | | | | | |
| GSM8K | 77.1 | 77.9 (+0.8) | 1.63× | 75.3 | 75.2 (-0.1) | 1.71× |
| GPQA | 25.2 | 25.7 (+0.5) | 1.82× | 27.0 | 26.6 (-0.4) | 1.66× |
| *Code Generation* | | | | | | |
| HumanEval | 30.5 | 30.5 (0.0) | 1.20× | 54.9 | 55.5 (+0.6) | 1.44× |
| MBPP | 37.6 | 37.4 (-0.2) | 1.35× | 54.0 | 54.6 (+0.6) | 1.33× |
| *Planning Tasks* | | | | | | |
| Countdown | 15.3 | 15.3 (0.0) | 2.67× | 14.6 | 14.6 (0.0) | 2.37× |
| Sudoku | 35.0 | 38.0 (+3.0) | 2.46× | 89.0 | 89.0 (0.0) | 3.40× |

uses the standard diffusion decoding with the complete step budget of $T_{\max}$ and **Prophet** employs early commit decoding with dynamic threshold scheduling. The threshold parameters are set to $\tau_{\text{high}} = 7.5$, $\tau_{\text{mid}} = 5.0$, and $\tau_{\text{low}} = 2.5$, with transitions occurring at 33% and 67% of the decoding progress. These hyperparameters were selected through preliminary validation experiments.

Our evaluation spans four capability domains to comprehensively assess Prophet's effectiveness. For general reasoning, we use MMLU (Hendrycks et al., 2021), ARC-Challenge (Clark et al., 2018), HellaSwag (Zellers et al., 2019), TruthfulQA (Lin et al., 2021), WinoGrande (Sakaguchi et al., 2021), and PIQA (Bisk et al., 2020). Mathematical and scientific reasoning are evaluated through GSM8K (Cobbe et al., 2021) and GPQA (Rein et al., 2023). For code generation, we employ HumanEval (Chen et al., 2021) and MBPP (Austin et al., 2021b). Finally, planning capabilities are assessed using Countdown and Sudoku tasks (Gong et al., 2024). We follow the prompt in simple-evals for LLaDA and Dream, making the model reason step by step. Concretely, we set the generation length $L$ to 128 for general tasks, to 256 for GSM8K and GPQA, and to 512 for the code benchmarks. Unless otherwise noted, all baselines use a number of iterative steps equal to the specified generation length. All experiments employ greedy decoding to ensure deterministic and reproducible results.

## 5.2 MAIN RESULTS AND ANALYSIS

The results of our experiments are summarized in Table 1. Across the general reasoning tasks, Prophet demonstrates its ability to match or even exceed the performance of the full baseline. For example, using LLaDA-8B, Prophet achieves 54.0% on MMLU and 83.5% on ARC-C, both of which are statistically on par with the full step decoding. Interestingly, on HellaSwag, Prophet (70.9%) not only improves upon the full baseline (68.7%) but also the half baseline (70.5%), suggesting that early commit decoding can prevent the model from corrupting an already correct prediction in later, noisy refinement steps. Similarly, Dream-7B maintains competitive performance across benchmarks, with Prophet achieving 66.1% on MMLU compared to the full model's 67.6%—a minimal drop of 1.5% while delivering 2.47× speedup.

Prophet continues to prove its reliability in more complex reasoning tasks, including mathematics, science, and code generation. For the GSM8K dataset, Prophet with LLaDA-8B obtains an accuracy

of 77.9%, outperforming the baseline's 77.1%. This reliability also extends to code generation benchmarks. For instance, on HumanEval, Prophet perfectly matches the full baseline's score with LLaDA-8B (30.5%) and even slightly improves it with Dream-7B (55.5% vs. 54.9%). Notably, the acceleration on these intricate tasks (e.g., 1.20× on HumanEval) is more conservative compared to general reasoning. This demonstrates Prophet's adaptive nature: it dynamically allocates more denoising steps when a task demands further refinement, thereby preserving accuracy on complex problems. This reinforces Prophet's role as a "safe" acceleration method that avoids the pitfalls of premature, static termination.

In summary, our empirical results strongly support the central hypothesis of this work: DLMs often determine the correct answer long before the final decoding step. Prophet successfully capitalizes on this phenomenon by dynamically monitoring the model's predictive confidence. It terminates the iterative refinement process as soon as the answer has stabilized, thereby achieving significant computational savings with negligible, and in some cases even positive, impact on task performance. This stands in stark contrast to static truncation methods, which risk cutting off the decoding process prematurely and harming accuracy. Prophet thus provides a robust and model-agnostic solution to accelerate DLM inference, enhancing its practicality for real-world deployment.

## 5.3 COMPARISON WITH OTHER ACCELERATION METHODS

We compare Prophet with two DLM acceleration methods, demonstrating that Prophet is orthogonal to and compatible with both.

**Comparison with distillation-based acceleration.** We compare against Self-Distillation Through Time (Deschenaux & Gulcehre, 2025) (SDTT), a training-based method that reduces the number of inference steps. We implement a preliminary version of SDTT on LLaDA-8B-Instruct, distilling a standard 256-step teacher into a 128-step student model (2× acceleration) and evaluate on GSM8K. As shown in Table 2a, SDTT effectively maintains accuracy (76.9%) while halving inference steps. Prophet Prophet achieves a competitive 1.63× speedup with slightly higher accuracy (77.9%), without any retraining cost. Most importantly, our results demonstrate that Prophet and SDTT are orthogonal and complementary. Crucially, Prophet and SDTT are complementary: applying Prophet's early-exit strategy on top of the SDTT-distilled student (Row 4) yields a 3.21× speedup with negligible accuracy drop, demonstrating that distilled models retain the early answer convergence property that Prophet exploits.

**Comparison with KV cache-based acceleration.** We compare against Fast-dLLM (Wu et al., 2026), a training-free method that reduces per-step computation via approximate KV caching and parallel decoding. We apply Prophet's early-exit mechanism on top of Fast-dLLM using LLaDA-8B on GSM8K. As shown in Table 2b, while Fast-dLLM alone achieves a 6.82× speedup by reducing per-step cost, combining it with Prophet yields a 7.66× total speedup without quality degradation. This multiplicative effect arises because the two methods operate on orthogonal dimensions: Fast-dLLM reduces the cost per step, while Prophet reduces the total number of steps required.

Table 2: Comparison between Prophet and acceleration baselines on GSM8K.

(a) Comparison with SDTT.

| Method | Accuracy (%) | Steps (Avg) | Speedup |
|---|---|---|---|
| LLaDA (Teacher) | 77.1 | 256 | 1.00× |
| SDTT (Distilled) | 76.9 | 128 | 2.00× |
| Prophet (Ours) | **77.9** | 160 | 1.63× |
| SDTT + Prophet | 76.4 | **79** | **3.21×** |

(b) Comparison with Fast-dLLM.

| Method | Accuracy (%) | Speedup |
|---|---|---|
| Baseline (LLaDA-8B) | 77.1 | 1.00× |
| Fast-dLLM (KV Cache + Parallel) | 76.6 | 6.82× |
| Prophet (Ours) | **77.9** | 1.63× |
| Fast-dLLM + Prophet | 77.3 | **7.66×** |

## 5.4 ABLATION STUDIES

Beyond the coarse step budget ablation above, we further dissect why Prophet outperforms static truncation by examining (i) sensitivity to the generation length $L$ and available step budget, (ii) compatibility with different remasking heuristics, and (iii) robustness to the granularity of semi-autoregressive block updates. Together, these studies consistently show that Prophet's adaptive early-commit rule improves the compute–quality Pareto frontier, whereas static schedules either under-compute (hurting accuracy) or over-compute (wasting steps).

Table 3: **Ablation study on step budget and remasking strategy.**. (a) Accuracy vs. step budget under two generation lengths $L$. Prophet stops early (average steps in parentheses) yet matches/exceeds the full-budget baseline. (b) Accuracy under different remasking strategies; Prophet complements token-selection policies.

(a) Accuracy vs. step budget and generation length.

| | Step Budget ($T_{\max}$) | | | | | |
|---|---|---|---|---|---|---|
| $L$ | 16 | 32 | 64 | 128 | Prophet (Avg Steps) | Full |
| 256 | 7.7 | 22.5 | 58.8 | 76.2 | **77.9** ($\approx$160) | 77.1 |
| 128 | 21.8 | 50.3 | 67.9 | 71.3 | **72.7** ($\approx$74) | 71.3 |

(b) Remasking strategy.

| Strategy | Baseline | Ours (Prophet) |
|---|---|---|
| Random | 63.8 | **66.6** |
| Low-confidence | 71.3 | **72.7** |
| Top-$k$ margin | 72.4 | **73.1** |

Table 4: **Sensitivity to block length** on GSM8K (semi-autoregressive updates). Prophet is less brittle to coarse-grained updates and yields larger gains as block length increases.

| Block length | 8 | 16 | 32 | 64 | 128 |
|---|---|---|---|---|---|
| Baseline | 67.1 | 68.7 | 71.3 | 59.9 | 33.1 |
| Ours (Prophet) | **72.8** | **73.3** | **72.7** | **69.8** | **52.2** |
| $\Delta$ (Abs.) | +5.7 | +4.6 | +1.4 | +9.9 | +19.1 |

**Accuracy vs. step budget under different $L$.** Table 3a summarizes GSM8K accuracy as we vary the number of refinement steps under two generation lengths ($L = 256$ and $L = 128$). Accuracy under a static step cap rises monotonically with more steps (e.g., $7.7\% \rightarrow 22.5\% \rightarrow 58.8\% \rightarrow 76.2\%$ for 16/32/64/128 at $L = 256$), but still underperforms either the full-budget decoding or Prophet. In contrast, Prophet stops adaptively at $\approx 160$ steps for $L = 256$ (saving $\approx 38\%$ steps; $256/160 \approx 1.63\times$) and yields a higher score than the 256-step baseline (77.9% vs. 77.1%). When the target length is shorter ($L = 128$), Prophet again surpasses the 128-step baseline (72.7% vs. 71.3%) while using only $\approx 74$ steps (saving $\approx 42\%$; $128/74 \approx 1.73\times$). These results reaffirm that the gains are not a byproduct of simply using fewer steps: Prophet avoids late-stage over-refinement when the answer has already stabilized, while still allocating extra iterations when needed.

**Remasking strategy compatibility.** Table 3b evaluates three off-the-shelf remasking heuristics (random, low-confidence, top-$k$ margin). Prophet consistently outperforms their static counterparts, with the largest gain under random remasking (+2.8 points), aligning with our earlier observation that random schedules accentuate early answer convergence. The improvement persists under more informed heuristics (low-confidence: +1.4; top-$k$ margin: +0.7), indicating that Prophet's stopping rule complements, rather than replaces, token-selection policies.

**Granularity of semi-autoregressive refinement (block length).** Table 4 shows that static block schedules are brittle: accuracy peaks around moderate blocks and collapses for large blocks (e.g., 59.9 at 64 and 33.1 at 128). Prophet markedly attenuates this brittleness, delivering consistent gains across the entire range, and especially at large blocks where over-aggressive parallel updates inject more noise. For instance, at block length 64 and 128, Prophet improves accuracy by +9.9 and +19.1 points, respectively. This robustness is a direct consequence of Prophet's time-varying risk-aversion: when coarse-grained updates raise uncertainty, the threshold schedule defers early commit; once predictions settle, Prophet exits promptly to avoid additional noisy revisions.

## 6 CONCLUSION

In this work, we identified and leveraged a fundamental yet overlooked property of diffusion language models: early answer convergence. Our analysis revealed that up to 99% of instances can be correctly decoded using only half of the refinement steps, challenging the necessity of conventional full-length decoding. Building on this observation, we introduced Prophet, a training-free early commit decoding paradigm that dynamically monitors confidence gaps to determine optimal termination points. Experiments on LLaDA-8B and Dream-7B demonstrate that Prophet achieves a reduction of up to $3.4\times$ reduction in decoding steps while maintaining generation quality. By recasting DLM decoding as an optimal stopping problem rather than a fixed-budget iteration, our work opens new avenues for efficient DLM inference in tasks with identifiable answer regions and suggests that early convergence is a core characteristic of how these models internally resolve uncertainty, across diverse tasks and settings.

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

APPENDIX

# A  DISCUSSION

**Scope and applicability.** Prophet is designed for tasks with identifiable answer regions, such as mathematical reasoning, code generation, and planning. In such settings, early answer convergence provides a reliable signal for termination. For open-ended generation tasks where the boundary between reasoning and answer is less distinct, applying Prophet presents non-trivial challenges, as the model may not exhibit clear convergence until late in the denoising trajectory. We therefore do not position Prophet as a universal accelerator for all text generation, but rather as a demonstration that early answer convergence is a practically exploitable property in structured generation settings.

**Conservative speedups on complex tasks.** Prophet exhibits more conservative speedups on complex tasks such as code generation ($1.20\times$ on HumanEval) compared to short-answer tasks ($3.40\times$ on Sudoku). This reflects an intended behavior: in tasks where the answer is semantically dependent on preceding reasoning chains, Prophet correctly detects persistent uncertainty and defers termination to preserve correctness. Combining Prophet with trajectory compression methods such as SDTT offers a natural solution, as distillation reduces the number of steps needed to form the reasoning chain, while Prophet further exploits convergence in the answer region. Our preliminary results confirm this, with SDTT + Prophet achieving a $3.21\times$ speedup on GSM8K compared to $1.63\times$ for Prophet alone.

**Learnable termination criteria.** The confidence gap metric used in Prophet is intentionally simple: it incurs negligible overhead and requires no additional training, making it easy to deploy. However, for tasks where model confidence does not reliably correlate with correctness, a lightweight learnable judge could provide more robust termination signals. We view a "Judge Prophet", which replaces the heuristic confidence gap with a trained discriminator, as a promising direction for future work, drawing on recent advances in judge-based decoding (Bachmann et al.).

**Integration with system-level optimizations.** As demonstrated in Section 5.3, Prophet is orthogonal to both distillation-based and cache-based acceleration methods. In KV Cache frameworks, Prophet's termination signal can be monitored by concatenating answer tokens with the active block, and once early commit is triggered, inference terminates immediately, saving the overhead of updating the cache for remaining steps. We consider this system-level synergy a compelling direction for future exploration.

# B  ADDITIONAL RESULTS

## B.1  CORROBORATING RESULTS ON EARLY ANSWER CONVERGENCE

This section presents additional results on MMLU complementing the GSM8K results in the main paper Section 3.2. We investigate two remasking strategies, low-confidence remasking and random remasking, and examine the effect of suffix prompting on the distribution of decoding steps at which correct answers first emerge.

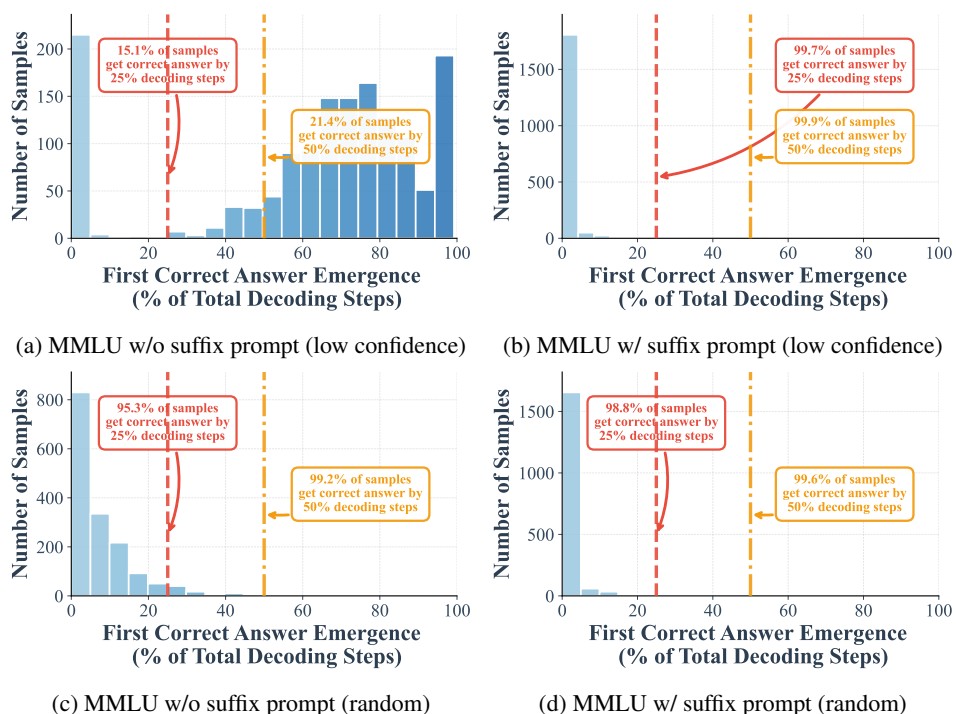

(a) MMLU w/o suffix prompt (low confidence)  (b) MMLU w/ suffix prompt (low confidence)

(c) MMLU w/o suffix prompt (random)  (d) MMLU w/ suffix prompt (random)

Figure 4: **Distribution of early correct answer detection during decoding process.** Histograms show when correct answers first emerge during diffusion decoding, measured as percentage of total decoding steps, using `LLaDA 8B` on MMLU. Red and orange dashed lines indicate 50% and 70% completion thresholds, with corresponding statistics showing substantial early convergence. Suffix prompting (b,d) dramatically accelerates convergence compared to standard prompting (a,c). This early convergence pattern demonstrates that correct answer tokens stabilize as top-1 candidates well before full decoding.

## B.2 QUALITATIVE ANALYSIS: DECODING TRAJECTORIES

To validate that DLMs identify answers before reasoning chains are complete, we present real decoding trajectories in Table 5. In the "Simple Arithmetic" case, the model locks onto the answer "3" at only 50% of steps, while the intermediate calculation "2/2 = 1" is still masked. This confirms the efficacy of Prophet's early commit mechanism. Our observation is that DLMs often internally converge on the final result significantly earlier than they finalize the justification or Reasoning Chain. Therefore, when Prophet triggers an early commit, the answer tokens are already stable and correct. While the intermediate CoT region might retain some noise or lack the final polish of a full-step generation, we have not observed significant degradations in the fluency or grammatical correctness of the final output.

Table 5: **Qualitative Analysis: Decoding Dynamics.** Visualization of the decoding trajectory for a simple arithmetic problem. Masked tokens are represented by MASK (note that sequences of consecutive masks are abbreviated for visual clarity). Crucially, even when the intermediate reasoning chain is incomplete, the model locks onto the correct final answer early in the process (highlighted in darker blue ).

| Stage | Decoded Sequence Stream (Case: Simple Arithmetic) |
|---|---|
| **Early**($\sim$10% steps) | To MASK MASK MASK add the MASK of blue MASK MASK the MASK of white MASK MASK MASK MASK 3 bolts . MASK MASK Answer : **The answer is 3** |
| **Prophet**($\sim$50% steps) | To find the total number of bolts , we need to add the MASK of blue fiber and the amount of white fiber . The robe takes 2 bolts of blue fiber . It also takes half that much white fiber , MASK MASK MASK 3 bolts . Final Answer : **The answer is 3** |
| **Full**(100% steps) | To find the total number of bolts , we need to add the amount of blue fiber and the amount of white fiber . The robe takes 2 bolts of blue fiber . It also takes half that much white fiber , so it takes 2 / 2 = 1 bolt of white fiber . To find the total number of bolts , we add the amount of blue fiber and the amount of white fiber : 2 ( blue fiber ) + 1 bolt ( white fiber ) = 3 bolts . Therefore , the robe takes a total of 3 bolts . Final Answer : **The answer is 3** |

### B.3 DISTRIBUTION OF INCORRECT ANSWERS

We analyzed the decoding dynamics of incorrect samples. We conducted an additional analysis tracking the "Last Change Step" for samples that eventually resulted in incorrect answers on GSM8K. As illustrated in Figure 5, updates for incorrect answers are heavily right-skewed (occurring in the last 20% of steps). This indicates that when the model produces an incorrect answer, it is typically uncertain rather than overconfidently wrong. It continues to fluctuate and flip its prediction until the very end of the generation process. This behavior serves as a natural safety mechanism for Prophet. Since incorrect answers rarely exhibit early stability or high confidence, they fail to trigger the "Early Commit" criteria. Consequently, Prophet correctly forces the model to utilize the full step budget for these difficult cases, rather than exiting prematurely. This explains why our method achieves significant speedups (by catching the left-skewed correct answers) without compromising accuracy (by avoiding the unstable incorrect ones).

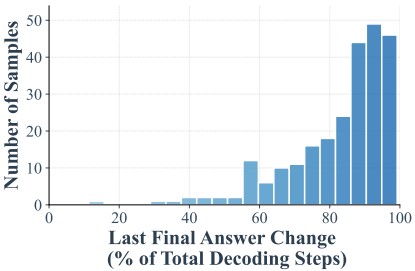

Figure 5: Distribution of the "Last Change Step" for incorrect answers. The right-skewed distribution ensures Prophet acts conservatively on uncertain samples.

## C IMPLEMENTATION DETAILS

### C.1 ALGORITHM DESIGN

**Suffix prompt mechanism.** The suffix prompt is inserted as a semantic anchor near the end of the generation window, followed by mask tokens reserved for the final answer. The resulting sequence structure is: `[Question] [MASKs for Reasoning Chain] The answer is [MASKs for Final Result]`. This mechanism accelerates convergence because diffusion

models generate bidirectionally. Without an explicit cue, the model spends early denoising steps implicitly determining output format. The suffix prompt eliminates this ambiguity by conditioning the model to expect a solution in the designated region, allowing answer tokens to stabilize significantly earlier and enabling Prophet to trigger early exit sooner, as illustrated in Figure 1 and Figure 2.

We note that utilizing a suffix prompt does not constitute the use of oracle information. The answer region position is deterministically assigned by the input prompt rather than derived from ground-truth labels. We instruct the model where to place the answer, not what the answer is. Both Prophet and the baseline use identical suffix prompt configurations; the performance gains therefore stem from convergence detection rather than any structural advantage.

**Answer region determination.** The answer region is defined based on the structural properties of each task:

- *Mathematical reasoning* (e.g., GSM8K): The dataset provides a standard separator between the reasoning chain and the final result. We define the answer region as the tokens following this separator.
- *General reasoning* (e.g., MMLU, ARC-C): These benchmarks consist of multiple-choice questions. The suffix prompt explicitly locates the position for the final prediction (e.g., "A" or "B"), and Prophet monitors the confidence gap over this single token.
- *Code generation* (e.g., HumanEval): The answer region is defined as the generated function body, delimited by a pre-inserted code block separator (e.g., ` `` `). The strict syntactic dependencies of code allow the model to converge on the correct program structure early, enabling safe acceleration without breaking syntax.

Regarding answer region length, we acknowledge that this specific setting relies on task priors. For example, in GSM8K, the answer is typically a concise number, whereas more complex mathematical problems (like AIME) might require variable-length outputs. However, this is not a fundamental limitation; in more complex scenarios, the fixed window can be replaced by dynamic semantic extraction techniques (similar to Wang et al. (2025a)) to identify the answer span on the fly. We opted for a pre-defined region primarily for implementation simplicity and conciseness. Ultimately, our objective is to reveal and rigorously analyze the intrinsic phenomenon of Early Answer Convergence, not to compete with engineering-oriented SOTA acceleration frameworks by building a universally applicable tool.

## C.2   EVALUATION AND HYPERPARAMETER

We re-implemented the evaluation of LLADA and DREAM on all reported datasets. Following standard practice, we generate and extract the final answer rather than comparing log probabilities in multiple-choice settings, which can slightly lower scores on some benchmarks when the model fails to produce a response in the expected format. Full experimental configurations are summarized in Table 6.

**Confidence schedule design.** Early commitment is governed by the three-stage confidence schedule defined in Eq. 5, designed around a principle of time-varying risk aversion. In early denoising steps, predictions are noisy, so we enforce a high threshold $\tau_{\text{high}}$ to prevent premature commitment. As decoding progresses and the marginal gain of further computation decreases, we progressively lower the threshold to encourage earlier exit.

**Hyperparameter robustness.** We did not perform an extensive hyperparameter search. A light sweep on a small GSM8K validation set revealed that three uniform stages with the thresholds in Table 6 work consistently well across all tasks. The robustness of this fixed configuration suggests that the confidence gap is a model-agnostic metric that generalizes without sensitive per-task tuning.

We chose a staged schedule for its simplicity and ease of implementation. To verify that our results are not contingent on this specific functional form, we conduct an ablation comparing the staged schedule against a continuous linear decay $\tau(p) = 8.0 - 4.5 \times p$ covering the same threshold range $[8.0, 3.5]$. As shown in Table 7, the two schedules yield comparable accuracy and speedup, confirming that the observed gains stem from the inherent early convergence property of the model rather than from a carefully engineered controller. Continuous or learnable schedules remain a promising direction for future optimization.

Table 6: **Configurations used in our runs.** We keep only parameters relevant to our method: base budget $(L, T, B)$ and PROPHET's confidence schedule defined in Eq. 5

| Benchmark | Base Budget $(L, T, B)$ | PROPHET Thresholds $(\tau_{\text{high}}, \tau_{\text{mid}}, \tau_{\text{low}})$ | Transition Points |
|---|---|---|---|
| MMLU | $L$=64, $T$=64, $B$=16 | (7.5, 5.0, 2.5) | 33%, 67% |
| ARC-C | $L$=64, $T$=64, $B$=16 | (7.5, 5.0, 2.5) | 33%, 67% |
| Hellaswag | $L$=64, $T$=64, $B$=16 | (7.5, 5.0, 2.5) | 33%, 67% |
| TruthfulQA | $L$=64, $T$=64, $B$=16 | (7.5, 5.0, 2.5) | 33%, 67% |
| WinoGrande | $L$=64, $T$=64, $B$=16 | (7.5, 5.0, 2.5) | 33%, 67% |
| PIQA | $L$=64, $T$=64, $B$=16 | (7.5, 5.0, 2.5) | 33%, 67% |
| GSM8K | $L$=256, $T$=256, $B$=32 | (8.0, 5.0, 3.5) | 33%, 67% |
| GPQA | $L$=256, $T$=256, $B$=32 | (8.0, 5.0, 3.5) | 33%, 67% |
| HumanEval | $L$=512, $T$=512, $B$=32 | (7.5, 5.0, 4.5) | 33%, 67% |
| MBPP | $L$=512, $T$=512, $B$=32 | (7.5, 5.0, 4.5) | 33%, 67% |
| Sudoku | $L$=24, $T$=24, $B$=24 | (7.5, 5.0, 2.5) | 33%, 67% |
| Countdown | $L$=32, $T$=32, $B$=32 | (7.5, 5.0, 2.5) | 33%, 67% |

Table 7: Comparison of threshold schedules on GSM8K (LLaDA-8B).

| Schedule Type | Accuracy (%) | Avg Steps |
|---|---|---|
| Staged (Ours) | 77.9 | 160 |
| Linear (Ablation) | 77.4 | 154 |

