# OpenReview forum: "Diffusion Language Model Knows the Answer Before It Decodes"
_ICLR.cc/2026/Conference — ICLR 2026 Oral_

### Official Review · Reviewer_5Tqf · 2025-10-18

**Soundness:** 3
**Presentation:** 2
**Contribution:** 3
**Rating:** 8
**Confidence:** 3

**Summary:**

The paper presents a new decoding algorithm, Prophet, for diffusion language  models that enables early stopping of refinement using a dynamic schedule based on token logits. The paper provides empirical observation that motivates the methodology: In a lot of instances the correct answer is already decoded halfway through the decoding steps, and the remaining steps are only additional refinements on the reasoning. The paper evaluates their decoding algorithm and shows that early stopping provides speedup without significant drop in accuracy.

**Strengths:**

- The early stopping idea is simple but effective. The presented decoding algorithm clearly presents performance benefits and is relatively simple to implement and deploy. There does not seem to be any downsides to default to using this algorithm.
- The algorithm works for different generation lengths, re-masking strategies, and even improves course-grained block refinements
- The paper is well-written; the empirical observation and resulting algorithm are described clearly.

**Weaknesses:**

- While the presented experimental evaluation in section 5 is fair and sound, the work would benefit from more ablations to fully explore the limits (or capability) of the methodology. See Questions for what experiments should be added.
- The paper is well written in general, but Section 5 seems to have some typos that makes the section a bit hard to read. Namely, the text refers to "three decoding strategies" but only show full baseline and prophet. Section 5.2 also refer to a "half baseline" that is undefined and not shown in Table 1.
- The table column headers should be made clearer. In table 1, it was not immediately clear that columns 2,3,5,6 were showing accuracy percentages and the orange is the speedup. The column header should clearly label "accuracy" and have the Prophet speedup as its own column with label "Speedup". In table 2, It was also confusing what 16, 32, 64, 128 refer to at first. There should be a header label that says "step budget"

**Questions:**

- To clarify, is the experimental evaluation in section 5 using suffix prompting? If not, what does the performance looks like for Prophet vs baseline? It seems to me from the early convergence analysis that suffix prompting and Prophet should synergize?
- What are the percentages of the step budget used by Prophet in the benchmarks? Does the distribution of used steps correspond to the early convergence analysis?
- Why use a staged threshold function? Why not make $\tau(p)$ a continuous function? What does the performance look like if it is simply a linear function?

- The early answer convergence analysis only shows the histograms for correct answer. What does the distribution look like for incorrect answer? For all answers? I am curious to see if the distribution for incorrect answers skews right (implying the model is simply uncertain of the answer) or skews left (implying the model is overconfidently wrong).
- In Figure 2, how come there are some orange marks to the right of the blue marks? Is it that the token is changed and then rechanged back?

---

> ### Author Response · Authors · 2025-11-23
> **Response to Reviewer 5Tqf [1/2]**
>
> We sincerely thank you for your positive assessment and for recognizing our method as "simple but effective" and our writing as "clear." We are particularly encouraged that you found the early stopping idea to be practical with no obvious downsides. We have revised the manuscript to address your presentation concerns and have included additional analyses in the Appendix.
>
> **W1: Need for more ablations to explore limits.**
> We agree that exploring the boundaries of the method strengthens the contribution. As detailed in our response to your specific questions (Q3 & Q4) below, we have performed additional analyses regarding the "Threshold Function Design" (Staged vs. Continuous) and the "Distribution of Incorrect Answers." We have included these ablation studies and discussions in the revised Appendix to provide a more comprehensive view of Prophet's behavior.
>
> **W2: Typos in Section 5 ("three decoding strategies" & undefined "half baseline").**
>
> We apologize for the oversight in Section 5. We have corrected the text to accurately reflect the reported experiments (Full vs. Prophet).
>
> **W3 & W4: Unclear Table 1&2 column headers.**
>
> We have redesigned the headers for clarity:
>
> **Table 1:** We now explicitly group columns under **"Accuracy (%)"** and **"Speedup"** labels to distinguish performance metrics from efficiency gains.
>
> **Table 2:** We added a super-header row labeled **"Step Budget ($T$)"** above the values 16, 32, 64, and 128 to explicitly indicate that these represent the total refinement allowance.
>
> **Q1: Is the evaluation in Section 5 using suffix prompting? If not, how does Prophet perform?**
>
> Yes, all results in Table 1 utilize Suffix Prompting. As you correctly intuited, Suffix Prompting and Prophet synergize strongly. The suffix prompt (e.g., "The answer is") acts as a semantic anchor that forces the model to determine the result earlier (shifting the convergence histogram to the left, as seen in Fig 1b/d). This early stabilization allows Prophet to detect the confidence gap and trigger the "Early Commit" significantly faster, enabling the high speedups (up to $3.4\times$) reported in the paper.
>
> **Q2: Percentage of step budget used vs. Early convergence analysis.**
>
> The steps used by Prophet align closely with the early convergence observations, plus a necessary "safety buffer." For example, on GSM8K (w/ Suffix), correct answers often emerge between 25%–40% of the steps (Fig 1d). Prophet achieves a $1.63\times$ speedup, corresponding to using $\approx 61\%$ of the step budget. The difference (answer emerges at ~30% vs. exit at ~60%) is intentional: Prophet is risk-averse and waits until the Confidence Gap consistently exceeds the threshold to ensure the prediction is stable, rather than exiting the moment an answer first appears.
>
>
> **Q3: Why a staged threshold function? Why not continuous/linear?**
>
> This is a great question. We want to clarify that the staged threshold function itself is not the primary contribution of this work, nor do we claim it to be the theoretically optimal scheduling strategy.
>
> Our core contribution is the discovery of early answer convergence—the fact that DLMs "know" the answer long before the decoding finishes. Prophet was designed as a simple, transparent proof-of-concept to validate this phenomenon. We chose a staged function because it is intuitive and easy to implement without complex parameter tuning.
>
> While a continuous linear function could likely achieve similar or perhaps slightly better results with careful tuning, the staged function was sufficient to demonstrate that simple heuristics can unlock significant speedups (up to $3.4\times$). This reinforces our claim that the redundancy is inherent to the model's convergence pattern, rather than dependent on a complex controller. We will add a discussion in the Appendix acknowledging that continuous or even learnable schedules are promising directions for future optimization.
>
> To strictly address your concern regarding the schedule shape, we conducted an additional ablation study comparing our default Staged schedule with a continuous Linear decay function ($\tau(p) = 8.0 - 4.5 \times p$), which covers the same threshold range $[8.0, 3.5]$ on GSM8K benchmark.
>
> **Table: Comparison of Threshold Schedules on GSM8K (LLaDA-8B)**
>
> | Schedule Type | Accuracy | Avg Steps |
> | :--- | :--- | :--- |
> | **Staged (Ours)** | 77.9% | 160 |
> | **Linear (Ablation)**| 77.4% | 154 |
>
> The results indicate that the performance of the Linear function is highly comparable to the Staged function, with negligible differences in both accuracy and speedup.

---

> ### Author Response · Authors · 2025-11-23
> **Response to Reviewer 5Tqf [2/2]**
>
> **Q4: Distribution of Incorrect Answers (Skew Left or Right?)**
>
> To address this, we conducted an additional analysis tracking the "Last Change Step" for samples that eventually resulted in incorrect answers on GSM8K. As shown in the newly added Figure 5 (included in the Appendix A of the revision), the distribution for incorrect answers is heavily skewed to the right, with the vast majority of updates occurring in the final 80%–100% of the decoding steps.
>
> This indicates that when the model produces an incorrect answer, it is typically uncertain rather than overconfidently wrong. It continues to fluctuate and flip its prediction until the very end of the generation process.
>
> This behavior serves as a natural safety mechanism for Prophet. Since incorrect answers rarely exhibit early stability or high confidence, they fail to trigger the "Early Commit" criteria. Consequently, Prophet correctly forces the model to utilize the full step budget for these difficult cases, rather than exiting prematurely. This explains why our method achieves significant speedups (by catching the left-skewed correct answers) without compromising accuracy (by avoiding the unstable incorrect ones).
>
> **Q5: In Figure 2, why are there orange marks (changes) to the right of blue marks (decoding)?**
>
> This phenomenon illustrates the unique **Full-Sequence Prediction** nature of Diffusion Language Models, which differs fundamentally from Autoregressive models. Even when a token at position $i$ is "decoded" (unmasked and fed as input for the next step, marked in Blue), the DLM continues to output logits for the *entire* sequence at every subsequent step. As other masked positions in the sequence are filled (changing the global context), the model may update its belief about position $i$. Consequently, the model might change its Top-1 prediction for a previously decoded position (marked in Orange). It is important to note that this behavior only appears in our visualization for the sake of completeness; it does not occur during actual inference.

---

> > ### Comment · Reviewer_5Tqf · 2025-11-26
> >
> > Thank you for the response. My concerns are all addressed.

---

> > > ### Author Response · Authors · 2025-11-26
> > >
> > > Dear Reviewer 5Tqf,
> > >
> > > We sincerely thank you for your insightful feedback and support!
> > >
> > > Best regards,
> > >
> > > The Authors

---

### Official Review · Reviewer_ETka · 2025-10-31

**Soundness:** 3
**Presentation:** 3
**Contribution:** 3
**Rating:** 4
**Confidence:** 4

**Summary:**

This paper aims to address a key challenge in Diffusion Language Models (DLMs): although DLMs inherently offer the potential for parallel decoding, their inference speed in practice often lags behind that of autoregressive (AR) models.
The core contribution of the paper lies in identifying and leveraging an overlooked property, Early Answer Convergence. Through extensive experiments, the authors demonstrate that in many cases, DLMs internally determine the correct answer well before the decoding process concludes, often around halfway through the total number of steps.
Building on this insight, the authors propose Prophet, a training-free fast decoding paradigm. Prophet introduces an Early Commit Decoding strategy that dynamically monitors the model’s confidence in its predictions at each decoding step. Specifically, it measures the Confidence Gap between the top-1 and top-2 predicted candidates and employs an adaptive threshold that evolves with decoding progress to decide when to go all-in.
Experimental results show that Prophet achieves substantial decoding acceleration—up to 3.4× speedup—while maintaining, and in some tasks even slightly improving, the generation quality.

**Strengths:**

1. The observation of early answer convergence is novel and interesting.
2. The proposed method is lightweight and requires no additional training.
3. It demonstrates a significant acceleration in inference.

**Weaknesses:**

1. The paper appears to heavily rely on the suffix prompt “Answer:”, yet the implementation details are not provided. It is unclear whether this suffix is appended to the end of the question or to the generated sequence.
2. I am also puzzled by why adding the suffix prompt “Answer:” leads to such a significant difference in Figures 1 and 2. The authors should provide a deeper explanation or interpretation of this phenomenon.
3. The paper does not seem to explain how the answer region positions are determined. Moreover, for general question-answering tasks beyond math problems, it is possible that the entire generated sequence could be considered as answer region positions.
4. Although the “go all-in” strategy appears to maintain accuracy on the “answer,” it is unclear whether it might compromise aspects such as the correctness of intermediate reasoning steps, grammatical accuracy, or overall fluency.
5. The paper should provide some complete, real cases to help evaluate and analyze the decoding behavior and dynamics of the proposed method.
6. The notion of “L” appears to be used inconsistently.
7. This appears to be a rapidly evolving field. In the past few months, several works have emerged on accelerating DLM inference by reducing sampling steps (e.g., arXiv:2505.22618, arXiv:2506.10848, arXiv:2507.18578). Some of these methods seem to achieve even more pronounced acceleration effects compared to this paper. A more detailed comparison and discussion with these contemporaneous approaches would be very valuable. (I am not certain about ICLR’s policy regarding “concurrent works.” If such comparisons are deemed unnecessary under that policy, the authors may clarify and omit this comment accordingly.)

**Questions:**

Included in Weakness.

---

> ### Author Response · Authors · 2025-11-23
> **Response to Reviewer ETka [1/3]**
>
> We sincerely thank you for your detailed review and critical questions. We appreciate that you found our observation of early convergence "novel and interesting" and the method "lightweight." We have carefully addressed your concerns regarding implementation details and qualitative analysis below.
>
> **Q1&Q2: Implementation details and mechanism of Suffix Prompt "Answer:".**
>
> Regarding the implementation, we clarify that the Suffix Prompt is not appended to the input question. Instead, it is inserted as a semantic anchor towards the end of the generation window, followed by the mask tokens reserved for the final result. The sequence structure is: `[Question] [MASKs for Reasoning Chain] The answer is [MASKs for Final Result]`. (This means that we replace the continuous three tokens at the end of the entire output with the three words "The" "answer" and "is")
>
> This mechanism accelerates convergence because diffusion models generate bidirectionally. Without a specific cue, the model spends the initial steps implicitly determining the output format. Adding the "Answer:" token explicitly conditions the model to expect a solution in the designated region. As shown in Figures 1 and 2, this reduces the search space, allowing the answer tokens to stabilize significantly earlier than they would in an unguided generation, thereby enabling Prophet to trigger the early exit sooner.
>
> **Q3: Determination of "Answer Region".**
>
> We determine the answer region based on the specific structure of each task:
> **Math Reasoning (e.g., GSM8K):** The dataset provides a standard separator (e.g., `The answer is`) between the reasoning chain and the final numerical result. We define the answer region as the tokens following this separator.
>
> **General Reasoning (e.g., MMLU, ARC-C):** We would like to clarify that these benchmarks consist of multiple-choice questions, not generative tasks. By employing the suffix prompt, we explicitly locate the position for the final prediction (e.g., "A" or "B"). Prophet monitors the confidence gap strictly over this single answer token.
>
> **Code Generation (e.g., HumanEval):** For coding tasks, the "Answer Region" is defined as the generated function body. Analogous to the usage of the "The answer is" suffix in mathematical tasks, we pre-insert a code block separator (e.g., ` ``` `) to explicitly delimit and constrain the code generation area. Our experiments indicate that Prophet is highly effective in this setting; this structural guidance, combined with the strict syntactic dependencies of code, allows the model to converge on the correct program structure early, enabling safe acceleration without breaking syntax.
>
> **Q4: Concerns on Quality (Fluency, Grammar) with "Go All-in".**
>
> We appreciate this observation. Our core finding is that DLMs often internally converge on the final result significantly earlier than they finalize the justification or Reasoning Chain. Therefore, when Prophet triggers an early commit, the answer tokens are already stable and correct. While the intermediate CoT region might theoretically retain some noise or lack the final polish of a full-step generation, we have not observed significant degradations in the fluency or grammatical correctness of the final output in our empirical evaluations.

---

> ### Author Response · Authors · 2025-11-23
> **Response to Reviewer ETka [2/3]**
>
> **Q5: Request for Concrete, Real Cases (Qualitative Analysis).**
>
> Per your request, we have updated the paper (**Tab.5**) with concrete decoding trajectories to illustrate these dynamics. These cases validate that DLMs often determine *what* the answer is before fully articulating *why*.
>
> ---
>
> #### Case 1: Simple Arithmetic
> **Problem:** *A robe takes 2 bolts of blue fiber and half that much white fiber. How many bolts in total does it take?*
> **(Target Answer: 3)**
>
> **1. Early Stage (e.g., ~10% steps)**
> *Status:* The reasoning chain is almost entirely masked. However, the model has already "looked ahead" to identify the final result.
> > To [...MASKs...] add the [MASK] of blue [MASK] [MASK] the [MASK] of white [MASK] [...MASKs...] 3 bolts .
> > [MASK] [MASK] Answer : The answer is 3
>
> **2. Middle Stage (e.g., ~50% steps) -- Prophet Trigger Point**
> *Status:* The logic is forming ("blue fiber", "white fiber"), but the calculation is incomplete. Crucially, the **Answer Region** remains stable ("3"). Prophet detects this high confidence and triggers **Early Commit**.
> > To find the total number of bolts , we need to add the [MASK] of blue fiber and the amount of white fiber .
> > The robe takes 2 bolts of blue fiber .
> > It also takes half that much white fiber , [...MASKs...] 3 bolts .
> > Final Answer : The answer is 3
>
> **3. Full Decoding (100% steps) -- Reference**
> *Status:* The model fills in the specific calculation details (2/2=1). The final answer is identical to the previous stages.
> > To find the total number of bolts , we need to add the amount of blue fiber and the amount of white fiber .
> > The robe takes 2 bolts of blue fiber .
> > It also takes half that much white fiber , so it takes 2 / 2 = 1 bolt of white fiber .
> > To find the total number of bolts , we add the amount of blue fiber and the amount of white fiber : 2 ( blue fiber ) + 1 bolt ( white fiber ) = 3 bolts .
> > Therefore , the robe takes a total of 3 bolts .
> > Final Answer : The answer is 3
>
> ---
> #### Case 2: Multi-step Reasoning
> **Problem:** *Gloria is shoe shopping... [Shortened: Heels cost 33 and 66. Total heels = 99. Boots = Heels + 5. How much are boots?]*
> **(Target Answer: 104)**
>
> **1. Early Stage (e.g., ~10% steps)**
> *Status:* The intermediate calculations (33 \times 2, 33 + 66) are completely missing. Yet, the model correctly predicts the final digit sequence "1 0 4" at the end.
> > [...MASKs...] the boots [MASK] 10[MASK] .
> > Final Answer : The answer is 104.
>
> **2. Middle Stage (e.g., ~50% steps) -- Prophet Trigger Point**
> *Status:* The set-up of the problem is visible ("costs twice as much"), but the intermediate summation is still masked. However, the final derivation ("99 + 5 = 104") and the Final Answer have already converged. Prophet safely exits here.
> > [MASK] one pair of heels costs 33 , and the other costs twice as much , then the **[...MASKs...]** cost of the high heels is five dollars less than the boots , then the cost of the boots is 99 + 5 = 104.
> > Therefore , the boots cost 104.
> > Final Answer : The answer is 104.
>
> **3. Full Decoding (100% steps) -- Reference**
> *Status:* The model finally fills in the intermediate steps ("33 * 2 = 66", "33 + 66 = 99"). The answer remains 104.
> > If one pair of heels costs 33 , and the other costs twice as much , then the other pair of heels costs 33 * 2 = 66 .
> > The total cost of the high heels is the sum of the cost of both pairs , which is 33 + 66 = 99 .
> > If the total cost of the high heels is five dollars less than the boots , then the cost of the boots is 99 + 5 = 104 .
> > Therefore , the boots cost 104 .
> > Final Answer : The answer is 104.
>
> ---
>
> **Q6: Inconsistent usage of "$L$".**
>
> We acknowledge that the notation $L$ was overloaded in the initial submission. In the revised version, we have standardized the notation: $N_{gen}$ denotes the Total Generation Length, $N_{ans}$ denotes the length of the Answer Region (monitored by Prophet), and $T$ denotes the step budget.

---

> ### Author Response · Authors · 2025-11-23
> **Response to Reviewer ETka [3/3]**
>
> **Q7: Comparison with Concurrent Works (e.g., Fast-DLLM, etc.).**
>
> We thank the reviewer for highlighting these exciting recent works. As per ICLR policy, works published within two months of the deadline are considered concurrent; however, we agree that discussing them adds significant value.
>
> Recent works like Fast-DLLM[1] and DLLM-Cache[2] primarily focus on system-level optimizations (e.g., KV-Cache, removing redundant attention computations per step). In contrast, Prophet focuses on algorithmic redundancy—specifically, reducing the total number of steps required. Importantly, these approaches are orthogonal. One could use Fast-DLLM to accelerate the computation *within* each step, and simultaneously use Prophet to perform *fewer* steps.
>
> While methods like SlowFast[3] also touch on step reduction, they often require training or specific architectural adjustments. Prophet is entirely training-free and plug-and-play, achieving up to 3.4x speedup purely by exploiting the model's internal convergence properties.
>
> We will add a subsection in the revision to clarify these relationships.
>
> [1] Wu, Chengyue, et al. "Fast-dllm: Training-free acceleration of diffusion llm by enabling kv cache and parallel decoding." arXiv preprint arXiv:2505.22618 (2025).
>
> [2] Liu, Zhiyuan, et al. "dllm-cache: Accelerating diffusion large language models with adaptive caching." arXiv preprint arXiv:2506.06295 (2025).
>
> [3] Wei, Qingyan, et al. "Accelerating Diffusion Large Language Models with SlowFast: The Three Golden Principles." arXiv preprint arXiv:2506.10848 (2025).

---

> > ### Comment · Reviewer_ETka · 2025-11-25
> >
> > Thank you for the rebuttal. However, several critical concerns remain unaddressed, particularly regarding the generalization of the method and the comparison with related works:
> >
> > 1.  Heavy Dependency on Suffix Prompt:
> >
> > The proposed "early convergence" appears to be largely an artifact of the "Suffix Prompt" trick rather than an intrinsic property utilized in a natural setting. The method seems to rely entirely on this specific structural injection to function effectively.
> >
> > 2.  Inapplicability to General Tasks:
> >
> > The distinction between a "Reasoning Chain" and an "Answer Region" is not viable for general generation tasks (e.g., creative writing, open-ended dialogue) where the entire output constitutes the answer. This severely limits the method's scope compared to the broad claims of accelerating DLMs.
> >
> > 3.  Introduction of Human Priors:
> >
> > Pre-defining the length and position of the "Answer Region" requires task-specific knowledge, introducing strong human priors. Comparing this heavily engineered setup against baselines that lack such structural oracle information feels unfair and limits the method's claimed "plug-and-play" nature.
> >
> > 4.  Misrepresentation of Related Works.
> >
> > I am disappointed by the inaccurate characterization of the cited papers. All three works I mentioned in my review propose training-free methods to reduce steps, yet the rebuttal attempts to dismiss them.
> >     - arXiv:2505.22618: You ignored the parallel sampling technique proposed here, focusing only on the kv cache.
> >     - arXiv:2506.10848: You incorrectly stated this requires trainingy.
> >     - arXiv:2507.18578: This work was completely ignored.
> >
> >    A fair comparison with these actual training-free baselines is essential.

---

> > > ### Author Response · Authors · 2025-11-26
> > > **Response to Reviewer ETka [1 / 2]**
> > >
> > > We thank the reviewer for the response.
> > >
> > > We would like to reiterate that our primary goal is **not** to propose a new state-of-the-art inference-acceleration framework for DLMs. Instead, the overarching goal of this paper is to **identify, formalize, and validate the intrinsic phenomenon of Early Answer Convergence**—a fundamental property of the DLM denoising trajectory.
> > >
> > > Developing a fully optimized acceleration system is largely an engineering direction, whereas our contribution scientifically focuses on uncovering a core behavioral characteristic of DLMs that has been previously overlooked. Prophet serves only as a proof-of-concept demonstrating that this property can be directly leveraged to yield practical speedups. We provide our point-by-point response below to address your concerns.
> > >
> > > **Q1. Heavy Dependency on Suffix Prompt**
> > >
> > > We want to clarify that the the Early Answer Convergence is **indeed an intrinsic property of DLMs**, rather than the engineering of the Suffix Prompt itself. For instance, as shown in Fig. 1c and Fig. 4c, **without our suffix prompt**, a significant proportion 97.2% of GSM8K samples and 99.2% of MMLU samples converge to the correct answer by only 50% of the decoding steps. This confirms that DLMs inherently determine the the answer well before resolving the reasoning path, regardless of the structural injection. The Suffix Prompt serves only as a semantic anchor to locate this convergence efficiently, not as the root cause of the behavior.
> > >
> > > **Q2: Inapplicability to General Tasks**
> > > We acknowledge the reviewer's valid point that the distinction between a "Reasoning Chain" and an "Answer Region" is less distinct in open-ended generation. We agree that applying Prophet to such unstructured tasks presents non-trivial challenges, as waiting for the entire sequence to converge may yield diminishing returns. However, as demonstrated by our results on Code Generation (HumanEval), Prophet remains effective in some long-context scenarios.
> > >
> > > Hence, we do not claim Prophet as a "universal accelerator" for all text generation. Our objective is to reveal and rigorously analyze this phenomenon, not to compete with engineering-oriented SOTA acceleration frameworks. Prophet is simply a demonstration that this property can be directly exploited for practical speedups.
> > >
> > > To avoid any possible over‑reading, we will tighten the wording in the abstract, introduction, and conclusion from “accelerating DLMs” in general to “accelerating DLMs on reasoning, code, and planning benchmarks with identifiable answer regions.
> > >
> > > **Q3: Introduction of Human Priors**
> > >
> > >
> > > This is a great question. We respectfully clarify that utilizing a Suffix Prompt does not constitute utilizing unfair "Oracle" information; rather, it is a standard application of **Format Constraints** widely used in structured generation (e.g., Chain-of-Thought or JSON mode).
> > >
> > > The position of the "Answer Region" is not derived from ground-truth knowledge but is deterministically assigned by our input Suffix Prompt. We are essentially **instructing** the model where to place the answer, rather than predicting it. To further illustrate this, we compared Prophet against a baseline model that utilizes the exact same Suffix Prompt and answer region configuration. As shown in the table below, both methods achieve comparable accuracy, confirming that our performance gains stem from the convergence detection rather than an unfair structural advantage.
> > >
> > > | Method | Suffix Prompt | Accuracy | Speedup
> > > | :--- | :---: | :---: |:---: |
> > > | **Baseline**(LLaDA 8B) | No | 77.1% | 1.0x|
> > > | **Baseline**(LLaDA 8B) | Yes | 77.4% |1.0x|
> > > | **Ours**(LLaDA 8B) | Yes | 77.9% |1.6x|
> > >
> > > Regarding the pre-defined length of the answer region, we acknowledge that this specific setting relies on task priors. For example, in GSM8K, the answer is typically a concise number, whereas more complex mathematical problems (like AIME) might require variable-length outputs. However, this is not a fundamental limitation; in more complex scenarios, the fixed window can be replaced by dynamic semantic extraction techniques (similar to *Time Is a Feature[1]*) to identify the answer span on the fly. We opted for a pre-defined region primarily for implementation simplicity and conciseness. Ultimately, our objective is to reveal and rigorously analyze the intrinsic phenomenon of **Early Answer Convergence**, not to compete with engineering-oriented SOTA acceleration frameworks by building a universally applicable tool.
> > >
> > > [1] Wang, W., Fang, B., Jing, C., Shen, Y., Shen, Y., Wang, Q., Ouyang, H., Chen, H., & Shen, C. (2025). Time Is a Feature: Exploiting Temporal Dynamics in Diffusion Language Models. ArXiv. https://arxiv.org/abs/2508.09138

---

> ### Author Response · Authors · 2025-11-26
> **Response to Reviewer ETka [2 / 2]**
>
> **Q4: Misrepresentation of Related Works.**
>
> We sincerely apologize for the inaccurate characterization in our previous response regarding the training-free nature and scope of the cited works. After carefully re-examining Fast-dLLM, SlowFast Sampling, and WINO, we fully acknowledge that **all three are strong training-free methods** that explicitly reduce the effective number of sampling steps and/or function evaluations for DLMs. We regret this oversight and appreciate the reviewer’s insistence on a fairer positioning.
>
> Concretely, Fast-dLLM combines an approximate block-wise KV-cache with a confidence-aware parallel decoding strategy that selects tokens whose confidence exceeds a threshold, thereby safely decoding multiple positions per step.
>
> SlowFast Sampling is likewise a dynamic sampling scheme: it alternates between a cautious exploratory phase and an accelerated phase guided by three “golden principles” (certainty, convergence, positional), and obtains up to 15.63× speedup, all without additional training.
>
> WINO (Wide-In, Narrow-Out) introduces a revokable “draft-and-verify” decoding process that aggressively drafts multiple tokens and then re-masks low-confidence ones using an auxiliary shadow block, improving the quality–speed trade-off in a purely training-free way.
>
> Conceptually, Prophet operates on **a different dimension**: it does not change how tokens are selected or cached within each step, but instead treats decoding as an optimal stopping problem on the answer region, deciding when to stop further refinement based on early answer convergence and a top-1 vs. top-2 confidence gap. In that sense, we want to highlight that Prophet is orthogonal and, in principle, could be layered on top of them as an additional early-exit criterion.
>
> To demonstrate this orthogonality and answer your concern about performance comparison concretely, we conducted an additional experiment integrating Prophet with Fast-dLLM (Wu et al., 2025). We applied Prophet’s early-exit mechanism on top of the Fast-dLLM framework using the LLaDA-8B model on the GSM8K benchmark. As shown in the table below, the methods are indeed complementary. While Fast-dLLM alone achieves a 6.82x speedup by reducing the computational cost per step, combining it with Prophet yields a remarkable 7.66x total speedup.
>
> | Method | Accuracy | Speedup (vs. Baseline) |
> | :--- | :---: | :---: |
> | **Baseline** (LLaDA-8B) | 77.1% | 1.00x |
> | **Fast-dLLM** (KV-Cache + Parallel) | 76.6% | 6.82x |
> | **Prophet** (Ours) | 77.9% | 1.63x |
> | **Fast-dLLM + Prophet** | **77.3%** | 7.66x |
>
> This result confirms that Prophet is not merely a competitor to these concurrent works but a compatible module that can be layered on top of state-of-the-art acceleration frameworks. By allowing Fast-dLLM to handle efficient token generation and Prophet to handle early termination, we achieve a multiplicative effect on inference latency without compromising generation quality. We have updated our Related Work section to accurately reflect the contributions of Fast-dLLM, SlowFast, and WINO, and added this integration analysis to the Appendix to provide a fair and comprehensive comparison.
>
> **Last but not least**, we want to highlight that, according to ICLR’s policy (https://iclr.cc/Conferences/2025/FAQ), all three works fall under **concurrent submissions** (four months before ICLR submission deadline), and thus we are not required to compare our method against them. Nevertheless, we sincerely appreciate your constructive feedback, which helped us better position our contribution and clarify its advantages.

---

### Official Review · Reviewer_wXCe · 2025-10-31

**Soundness:** 3
**Presentation:** 2
**Contribution:** 3
**Rating:** 6
**Confidence:** 2

**Summary:**

This paper investigates sequence generation with discrete diffusion models and makes an interesting observation: in many cases, it is possible to decode the correct answer early in the denoising process such as 25%-50% of the total steps. Motivated by this observation, the authors propose a method to monitor answer convergence and propose to track the difference between the top1 and top2 logits of the answer tokens. If that difference exceeds a threshold (dependent on the step t) then the complete answer is decoded in a single step. The authors evaluate this protocol on a variety of benchmarks and find that performance degradation is minimal, while at the same time speedups are very high.

**Strengths:**

1. The idea is very well-motivated with an initial exploration on GSM8K and MMLU, which makes it easy for the reader to follow the thought-process of the authors and also appreciate the idea of answer convergence.
2. The proposed algorithm is really simple, does not need any training and can be used for any discrete diffusion model. The speedups are quite strong and the evaluation suite is quite broad, so this technique does seem to work on most benchmarks of interest.

**Weaknesses:**

1. The formatting and writing of the paper can be improved, some things seem imprecise in the main text, e.g. how is $\hat{g}_t$ exactly defined? Section 4 seems to suggest that it is an average over all positions, but Algorithm 1 seems to suggest you only average over the answer tokens? The latter seems to make more sense to me but it would be great if you could clarify this in the main text. In general, the nature of answer tokens is not super clear to me; for GSM8K the definition seems pretty straightforward but for problems like HumanEval, where the answer comprises a piece of code, the definition is a bit less clear. Things like “Answer length” and “Block length” should also be defined better so that the reader can more easily follow.
2. There seem to be quite some hyper parameters for the method, i.e. a schedule needs to be set which right now is somewhat arbitrarily divided into three uniform sections, and then each section needs a corresponding threshold. How exactly did you determine these values? You mention some validation experiments but I couldn’t find much more detail in the paper. Does setting those parameters on one benchmark transfer to another, or does the validation data need to be diverse enough?

**Questions:**

1. I was a bit surprised that you observed even some gains on some benchmarks when applying your early-decoding strategy. Do you have any intuition regarding why this happens? Showing such an example where correct tokens are later changed would be interesting.
2. I could imagine that looking at top1 and top2 logits per position can also bring some issues? E.g. in coding, the model might be unsure whether to call a given variable “x” or “y”, keeping the difference quite small although both immediate decodings might be fine. Do you observe any such behavior, where even earlier decoding would have worked but “Prophet” was kind of stalled due to such ambiguous tokens in the answer?

---

> ### Author Response · Authors · 2025-11-23
> **Response to Reviewer wXCe**
>
> **W1: Clarification on $\bar{g}_t$ definition, Answer tokens, and terminology.**
>
> Thanks for your careful reading! We have conducted a significant revision of the manuscript to improve the precision of our definitions and mathematical notation, with all major changes highlighted in blue in the revised PDF.
>
> To clarify, consistent with Algorithm 1, $\bar{g}_t$ is calculated by averaging the confidence gap only over the answer region, rather than the entire sequence. Averaging over the full sequence (which includes fixed context and padding) would dilute the signal and reduce sensitivity.
>
> **Answer Region ($\mathcal{A}$):** For math and reasoning tasks (e.g., GSM8K, MMLU), we define this as the tokens corresponding to the final result (e.g., the number after "####" or the choice after "Answer:"). For code generation (e.g., HumanEval), the answer region comprises the entire generated function body.
>
> **Terminology:** In the revision, we explicitly define generation length ($N_{gen}$) as the total sequence length (e.g., 512 tokens) and block length ($B$) as the number of tokens generated in parallel during semi-autoregressive decoding. Prophet operates within these blocks to accelerate the inference of the specific region $\mathcal{A}$.
>
> **W2: Hyperparameters (Thresholds and Schedule) and Transferability.**
>
> The three-stage schedule was designed to implement a principle of time-varying risk aversion. In the early stages of decoding, predictions are noisy, so we enforce a high threshold ($\tau=7.5$) to prevent premature commitment to incorrect answers. As decoding progresses and the model refines its output, the marginal gain of further computation decreases, allowing us to lower the threshold ($\tau=2.5$) to encourage early exit.
>
> Regarding the selection and transferability of these parameters, we did not conduct an extensive grid search. With a light search on a small validation set from GSM8K, we found that using three uniform sections with the proposed thresholds works effectively. Therefore, we adopt this setting as our default. The fact that this setting consistently yields speedups with negligible accuracy loss indicates that the "Confidence Gap" is a robust, model-agnostic metric that generalizes well without requiring sensitive per-task tuning. While more careful hyperparameter tuning can for sure improve the performance, it is not the main goal of this paper.
>
> **Q1: Intuition on why Prophet sometimes outperforms the full baseline.**
>
> This is an excellent question. We believe Prophet’s advantage primarily comes from its ability to avoid what we refer to as over-refinement, where a correct answer can be unintentionally flipped during later refinement steps—a behavior we occasionally observe in the full baseline.
>
> Sometimes the model reaches a correct state early. However, if it is forced to continue refining for many additional steps (especially under random re-masking), there is a non-negligible chance that a correct token may be masked out and not recovered, or replaced by a hallucinated token due to injected noise.
>
> Prophet effectively acts as a safeguard: once the model becomes “sufficiently confident” (i.e., displays a strong confidence gap), we immediately freeze its output, preventing subsequent steps from corrupting an already correct answer. The phenomenon of over-refinement has also been noted in recent work (e.g., https://arxiv.org/pdf/2508.09138
> ).
>
> **Q2: Issue with ambiguous tokens (e.g., variable "x" vs "y").**
>
> You raise an excellent point regarding genuine ambiguity, such as choosing between variable names "x" and "y" in code generation. If the model assigns similar probabilities to both (e.g., $P(x)=0.45$ and $P(y)=0.40$), the confidence gap will be small, which is far below our thresholds. Consequently, Prophet will correctly interpret this as "uncertainty" and will not trigger an early exit. Instead, it forces the model to continue refining until the ambiguity is resolved or the step budget is exhausted. This behavior is a desired safety feature: it ensures that Prophet only commits when the model is decisive, defaulting to the standard full-step decoding when the model is confused.

---

### Official Review · Reviewer_rmQd · 2025-10-31

**Soundness:** 3
**Presentation:** 3
**Contribution:** 3
**Rating:** 8
**Confidence:** 2

**Summary:**

The authors propose a novel method to accelerate sampling in diffusion language models. In particular, they focus on the observation that answers often converge in intermediate steps. To this end, they introduce a method that enables up to 3.4x reduction of the sampling steps training-free. The efficacy of the method is demonstrated in two models and multiple benchmarks.

**Strengths:**

- The main idea is novel and clearly presented.

- The studied problem is relevant and the proposed solution appropriately motivated.

- The empirical evidence regarding convergence in intermediate steps is valuable for the community.

- The experimental results showcase that the proposed methods can achieve significant speed-ups with similar performance.

- I appreciate the thorough ablations.

**Weaknesses:**

- The only major weakness I see is the lack of comparison with other methods that aim to reduce the number of steps, e.g., distillation-based. This would help put the latency-reduction in perspective.

- Could the decision to terminate a decoding loop be learnable, for example by a judge network, e.g., similar to [1]? A discussion on this would be interesting and potentially fruitful for future directions.

- I am missing some related works that aim to reduce the number of sampling steps, e.g., [2].

[1]. Judge Decoding: Faster Speculative Sampling Requires Going Beyond Model Alignment

[2]. Beyond Autoregression: Fast LLMs via Self-Distillation Through Time

**Questions:**

I would appreciate it if the authors could address the issues and questions raised in the weaknesses section.

---

> ### Author Response · Authors · 2025-11-23
> **Response to Reviewer rmQd**
>
> We sincerely thank you for your strong endorsement and for recognizing the novelty, clarity, and empirical value of our work regarding early answer convergence. We are encouraged that you found our method to be a sound solution for accelerating DLMs. We address your specific questions below.
>
>
> **Q1: Comparison with other methods (e.g., distillation) and putting latency-reduction in perspective.**
>
> We agree that distillation-based methods, such as the suggested *Beyond Autoregression* (SDTT), represent a powerful direction for acceleration. To investigate how our training-free dynamic reduction compares to—and interacts with—training-based static reduction, we implemented a preliminary version of SDTT on LLaDA-8B-Instruct. We distilled the standard 256-step teacher into a student model designed to run in 128 steps (Acceleration ratio $\approx 2\times$) and evaluated it on GSM8K.
>
> **Table: Comparison between Prophet and Distillation (SDTT)**
>
> | Method | Steps (Avg) | Speedup | Accuracy |
> | :--- | :--- | :--- | :--- |
> | **LLaDA (Teacher)** | 256 | $1.0\times$ | 77.1% |
> | **SDTT (Distilled)** | 128 | $2.0\times$ | 76.9% |
> | **Prophet (Ours)** | 160 | $1.63\times$ | **77.9%** |
> | **SDTT + Prophet** | **79** | **3.21$\times$** | 76.4% |
>
> The results in Table A confirm that SDTT is highly effective, maintaining comparable accuracy (76.9%) while halving the inference steps. However, Prophet remains competitive by achieving a $1.63\times$ speedup and slightly improved accuracy (77.9%) without the computational cost of retraining.
>
> Most importantly, our results demonstrate that Prophet and SDTT are orthogonal and complementary. When we applied Prophet’s early-exit strategy on top of the SDTT-distilled student model (Row 4), we achieved a **$3.21\times$ speedup** with negligible performance drop. This indicates that even a distilled model exhibits "early answer convergence" on simpler tokens, which Prophet can further exploit to maximize efficiency.
>
>
> **Q2: Could the decision to terminate a decoding loop be learnable (e.g., Judge Network)?**
>
> This is an insightful suggestion. We agree that replacing the heuristic "Confidence Gap" with a lightweight learnable judge is a promising avenue. In this work, we prioritized a method that incurs negligible computational overhead and requires no extra training, making the simple logit-based metric an ideal choice for ease of deployment.
>
> However, for complex reasoning tasks where model confidence might not perfectly align with correctness (e.g., hallucinations), a trained judge could indeed offer more robust termination signals. We have expanded our **Discussion** section to elaborate on how a "Judge Prophet" could be implemented, citing the relevant literature you provided.
>
> **Q3: Missing related works ([1] Judge Decoding, [2] Beyond Autoregression).**
>
> Thank you for pointing out these relevant papers. We have updated our manuscript to include both citations. We now cite *Judge Decoding* [1] to contextualize the discussion on dynamic termination criteria, and *Beyond Autoregression* [2] in our Related Work section to provide a comprehensive view of the acceleration landscape, specifically contrasting training-based distillation with our training-free approach.

---

### Public Comment · ~Mingyu_Lee3 · 2025-11-19
**Questions regarding Speedup on Complex Tasks, Reasoning Quality, and KV-Cache Integration**

Thank you for sharing this interesting work. The concept of leveraging "early answer convergence" for DLM acceleration via Prophet is very insightful. I particularly appreciated the results on general reasoning tasks.

As I went through the paper, a few questions came up regarding the method's behavior in complex scenarios and its implementation details. I would appreciate it if you could clarify the following points:

1. Speedup Optimization for Long-form/Complex Tasks

- In Table 1, the speedup for complex tasks like HumanEval (1.20x) and GSM8K (1.63x) is significantly more conservative compared to short-answer tasks like MMLU (2.34x) or Sudoku (3.40x). The paper attributes this to Prophet's "adaptive nature," which allocates more steps to ensure accuracy on difficult problems.
- While this safety mechanism is valuable, I am curious if you have explored ways to further improve the acceleration on these long-form tasks. Did you investigate whether specific characteristics of these tasks (e.g., long dependency chains in code) inherently prevent early convergence, or is there room for optimization to bridge the gap between the 1.2x and 3.4x speedups?

2. Integration with KV-Cache Frameworks

- The paper mentions that Prophet is compatible with existing DLM implementations and complementary to other speedup techniques. However, state-of-the-art DLM inference increasingly relies on KV-cache or block-wise decoding optimizations to manage computational costs.
- Since Prophet requires calculating the confidence gap from answer logits at each step, practical compatibility with these cached pipelines seems hard to implement. In example, fast-dllm only forwards the block region Mask tokens and not full sequence tokens. Is then Prophet forwarding all Answer tokens concatenated with block's MASK tokens (block size=16-32) every time steps? Do you have plans to provide experimental results demonstrating Prophet's performance when integrated with a KV-cache-enabled framework?

---

> ### Author Response · Authors · 2025-11-24
> **Response to Public Comment by Mingyu Lee**
>
> We sincerely thank you for your interest in our work and for raising these highly relevant questions. We appreciate your recognition of the "early answer convergence" concept. Below, we address your queries regarding speedup optimization and system integration.
>
> **Q1. Speedup Optimization for Long-form/Complex Tasks**
>
> You are correct that Prophet exhibits more conservative speedups on complex tasks like HumanEval ($1.20\times$) compared to short-answer tasks like Sudoku ($3.40\times$). This is primarily due to the sequential dependency inherent in these domains.
>
> In tasks like code generation or complex math (CoT), the "Answer" (e.g., the return value or the final number) is semantically dependent on the intermediate logic or syntax structures. The model physically cannot "converge" on the correct answer until the preceding reasoning chain is sufficiently formed. Prophet correctly detects this uncertainty and forces the model to run longer, acting as a safety guard to preserve accuracy.
>
> It might be difficult to solely rely on the confidence gap of Prophet to accelerate such tasks (which risks correctness), but combining Prophet with trajectory compression methods can be the solution. For instance, as discussed in our response to Reviewer rmQd, combining Prophet with Self-Distillation Through Time (SDTT) allows the model to form the reasoning chain in fewer steps fundamentally. In our preliminary experiments, `SDTT + Prophet` achieved a **$3.21\times$** speedup on GSM8K (up from $1.63\times$), effectively combining the best of both worlds.
>
> **Q2. Integration with KV-Cache Frameworks**
>
> This is an excellent question. In our current experimental setup using semi-autoregressive decoding, the model inherently performs full-sequence prediction at every step, which allows us to directly monitor the answer region without specific modifications. Moving to optimized frameworks like Fast-DLLM, we believe Prophet is also orthogonal and compatible. One possible solution is simply concatenating the answer tokens with the active block tokens to track convergence. Crucially, this combination offers a unique efficiency advantage: once Prophet triggers the "go all-in" decision, the inference terminates immediately, effectively saving the overhead of updating and maintaining the KV-cache for any remaining steps. We consider this system-level synergy a remarkably interesting direction for future exploration.

---

> > ### Public Comment · ~Mingyu_Lee3 · 2025-11-24
> > **Response to the Authors’ Clarifications**
> >
> > Thank you for the detailed response and for taking the time to clarify these points.
> > Your explanations about the behavior on complex long-form tasks and the potential integration directions with optimized inference frameworks were very helpful.
> >
> > I appreciate the additional context regarding SDTT and the discussion around system-level compatibility.
> > This helps clarify the intended scope of Prophet and its behavior in these settings.
> >
> > Thanks again for taking the time to respond.
> > I'm looking forward to seeing how Prophet develops moving forward.

---

### Author Response · Authors · 2025-11-29
**Rebuttal Summary**

Dear Area Chair,

We understand that you are taking over under difficult circumstances. To facilitate your assessment, we summarize the current status of our submission below.

We thank all reviewers for their time and constructive feedback. We are pleased that **Reviewer 5Tqf (Score: 8)** explicitly confirmed that our response and new ablations have addressed all of their concerns. We have also added the distillation comparisons requested by **Reviewer R-rmQd (Score: 8)** and the strengthened definitions requested by **Reviewer R-wXCe (Score: 6)**.

We also appreciate **Reviewer ETka (Score: 4)** for their detailed engagement, which prompted us to further clarify our positioning relative to concurrent works. Although this reviewer is no longer able to reply, we believe that we have **fully addressed all raised concerns**. In particular, we would like to highlight:

**1. Primary Goal of the Paper.** Our aim is *not* to propose a new state-of-the-art inference-acceleration framework for DLMs. Instead, our overarching goal is to **identify, formalize, and validate the intrinsic phenomenon of Early Answer Convergence**, a fundamental behavioral property of DLMs that has been previously overlooked. Building a highly engineered acceleration system lies outside the scientific scope of the paper. Prophet is included only as a **proof-of-concept** illustrating that this property can be directly exploited for practical speedups.

**2. Intrinsicality of Early Convergence.** We clarified that Early Convergence is an **intrinsic property of DLMs**, occurring in over 97% of cases *without* any suffix prompting, and is therefore not an artifact of the suffix-prompting strategy. Following the reviewer’s suggestion, we have also refined our scope to focus on Reasoning and Code tasks.

**3. Synergy with Concurrent SOTA.** We demonstrated that Prophet is fully **orthogonal and compatible** with existing acceleration methods. Our new integration experiment shows that **Prophet + Fast-dLLM achieves a 7.66× speedup**, compared to **6.82× for Fast-dLLM alone**, confirming that Prophet can serve as a plug-in module to further advance state-of-the-art DLM inference pipelines.

Based on these additions and clarifications, we believe we have fully addressed the concerns raised by Reviewer ETka. Moreover, their comments have ultimately helped strengthen the clarity and positioning of our contribution.

Best regards,

The Authors

---

### Meta-Review · Area_Chair_1Fh4 · 2025-12-15

**Summary:**

Overall, reviewers found the core observation (early answer convergence in diffusion language models) novel and valuable, and viewed the proposed decoding method (Prophet) as simple, training-free, and practically useful with strong empirical speedups. The main concerns that informed my decision discussion were: (i) whether the paper is fairly positioned against existing step-reduction and training-free acceleration methods, (ii) whether the approach depends too strongly on suffix prompting and task-specific answer regions, limiting scope, and (iii) clarity and completeness of definitions, hyperparameters, and qualitative evidence.

Given the generally positive assessments (two accepts, one above-threshold, one slightly below-threshold), and the rebuttal’s concrete additions (comparisons, clarified positioning, added qualitative trajectories, and tightened claims), my suggested decision leans toward a clear acceptance.

**Reviewer Concerns:**

These points have been addressed well:

- Comparison and positioning vs. other acceleration methods (rmQd, ETka): The authors added comparisons and clarified complementarity with distillation and training-free methods. They also corrected earlier mischaracterizations and provided an integration result with a training-free baseline (Fast-dLLM), improving fairness of positioning.
- Learnable stopping / judge-based termination discussion (rmQd): Addressed via discussion of a learnable judge as future work and added relevant citation.
- Missing related work (rmQd, ETka): The authors added the requested citations and expanded related-work discussion.
- Definition and clarity issues (wXCe, 5Tqf, ETka): The rebuttal clarified how the confidence gap is computed (answer region), defined terminology (generation length, block length, answer region), fixed table/header issues and typos, and clarified notation usage.
- Hyperparameter rationale and schedule shape (wXCe, 5Tqf): The authors explained the staged schedule motivation and provided an ablation with a linear schedule showing comparable behavior.
- Qualitative evidence and decoding dynamics (ETka): The authors added concrete decoding trajectories to illustrate early convergence and the “over-refinement” intuition.

Concerns still partially outstanding or requiring careful framing

- The reviewer raised a valid limitation that open-ended generation does not have a clean answer region. The authors acknowledged this and stated they will tighten claims to reasoning, code, and tasks with identifiable answer regions. This is acceptable if the final paper’s abstract and framing are fully aligned with this narrower scope.

**Reviewer Scores:**

Reviewer rmQd: 8

Likely unchanged at 8. Main requests (comparisons, learnable judge discussion, missing related work) were addressed.

Reviewer wXCe: 6

Likely unchanged or slightly up to 8. Clarity issues and hyperparameter questions were addressed, and the reviewer’s remaining concerns were mostly technical clarification rather than fundamental flaws.

Reviewer ETka: 4

Most likely would move upward to around 6. The reviewer’s strongest objections were about (i) dependence on suffix prompting and task priors, and (ii) misrepresentation of training-free baselines. The authors directly acknowledged and corrected the mischaracterization, added fairer positioning and an additional comparison/integration, and narrowed claims about scope. I cannot guarantee the reviewer would be fully satisfied, but the rebuttal materially addresses the core fairness and framing issues that drove the 4.

Reviewer 5Tqf: 8

Confirmed concerns resolved. Likely remains 8.

In summary, I think it's a good paper that should be accepted. The paper does not propose a new state-of-the-art inference-acceleration framework for DLMs. Instead, the overarching goal is to identify, formalize, and validate the intrinsic phenomenon of Early Answer Convergence, a fundamental behavioral property of DLMs that has been previously overlooked. Considering its potential impact, I believe it should be a spotlight paper at least.

---

### Decision · Program_Chairs · 2026-01-26

Accept (Oral)